# REWARD DROPOUT IMPROVES CONTROL: BI-OBJECTIVE PERSPECTIVE ON REINFORCED LM

## ABSTRACT

We study the theoretical aspects of Reinforced Language Models (RLMs) from a bi-objective optimization perspective. Specifically, we consider the RLMs as a Pareto optimization problem that maximizes the two conflicting objectives, i.e., reward objective and likelihood objectives, simultaneously. Our main contribution consists of three parts. First, we establish the theoretical foundations of RLM as a Pareto optimization problem by presenting Reward Upper BOund (RUBO) and Pareto optimality. Our theoretical outcomes are supported by not only deductive proofs but also empirical results. Second, we propose Reward Dropout, a simple yet powerful method that guarantees to improve a bi-objective optimization of RLM. Lastly, we demonstrate that the Reward Dropout is consistently effective across five benchmark datasets and four benchmark LLMs, meaning that the Reward Dropout significantly improves the optimization performance of RLMs.

## 1 INTRODUCTION

The emergence of ChatGPT has sparked public interest in language models (LMs), resulting in a surge of LM research in both academia and industry. In particular, the use of reinforcement learning (RL) to control LMs has emerged as a significant research topic. In fact, leveraging RL to fine-tune LMs, or reinforced language models (RLMs) (Stiennon et al., 2020; Korbak et al., 2022; Ouyang et al., 2022; Bai et al., 2022), has long been studied as one of the general approaches to building controllable language models (CLMs) (Hu et al., 2017; Liu et al., 2022; Zhang et al., 2022a; Liu et al., 2023), where the goal is to generate sequences of intended attributes. The sequences here include texts (Yu et al., 2017; Li et al., 2017b; Ziegler et al., 2019; Liu et al., 2020a; Ouyang et al., 2022), melodies (Jaques et al., 2017; Jiang et al., 2020), molecules (Guimaraes et al., 2017; Olivecrona et al., 2017; Popova et al., 2018), menu lists (Chen et al., 2015; Lee et al., 2021; Mårtensson, 2021), purchase behaviors (Zhao et al., 2017; Bai et al., 2019; Zou et al., 2019; Shin et al., 2022), etc. Despite its long history and recent popularity, however, there is still a lack of theoretical understanding of how RLM works, under what conditions it succeeds or fails, and whether it can be guaranteed to improve control performance.

In this work, we study the theoretical aspects of RLMs through the lens of a Pareto optimization problem. Specifically, in Section 2, we consider the objective function of RLM as the off-policy RL problem (see Eq (4)) and in Section 3, we recast the RLM from a bi-objective problem that has the nature of a Pareto optimization (see Eq (6)). Section 4 presents theoretical and empirical evidence that the RLM is indeed a Pareto optimization problem. Based on this evidence, we propose Reward Dropout, a simple yet powerful method that guarantees to improve the bi-objective optimization of RLM. Finally, in Section 5, we evaluate the performance of Reward Dropout on five RLM benchmark datasets. Reward Dropout, which has its theoretical origins in Theorem 4.3, showed significant performance improvements on all RLM benchmark datasets. Our contributions are summarized as follows:

- Show that RLMs can be analyzed from a bi-objective perspective, which has the nature of Pareto optimization.

- Propose a simple yet powerful method named Reward Dropout that guarantees to improve the bi-objective optimization of RLMs.

- Demonstrate the effect of Reward Dropout is consistent across five benchmark datasets and four benchmark LLMs.

## 2 PRELIMINARIES

### 2.1 CONTROLLABLE LANGUAGE MODELS

Controllable language models (CLMs) are the models designed to address a controlled text generation (Hu et al., 2017; Liu et al., 2022; 2023; Zhang et al., 2022a). That is, the CLM aims to inject a specific control code $c$ into a language model (LM) so that the sequence (trajectory) $\tau = [x_1, x_2, ..., x_T]$ is generated as intended. Current approaches for modeling CLMs include the class conditional language model (CCLM) (Ficler & Goldberg, 2017; Dai et al., 2019; Keskar et al., 2019; Sudhakar et al., 2019), Bayesian controllable language model (BCLM) (Dathathri et al., 2019; Krause et al., 2020; Yang & Klein, 2021; Lu et al., 2021; Li et al., 2022), and reinforced language model (RLM) (Bian et al., 2019; Yu et al., 2017; Xu et al., 2018; Luo et al., 2019; Liu et al., 2020c; Stiennon et al., 2020; Korbak et al., 2022; Ouyang et al., 2022; Bai et al., 2022).

In CCLM approach, we prepend a code $c$ to the sequence, i.e., $\text{concat}(c, \tau)$, so that the language model parameters $\theta_{\text{lm}}$ are directly updated on $c$,

$$\ln p_{\text{clm}}(\tau|c) := \sum_{t=1}^{T} \ln p_{\text{lm}}(x_t|x_{<t}, c) . \tag{1}$$

Note that a target code $c'$ is fed to $p_{\text{clm}}(\hat{\tau}|c')$ to intend a specific control during inference. On the other hand, in BCLM approach, we separate the control part from the language model using Bayes' theorem by defining a distinctive classifier $p_{\text{cls}}(c|\tau)$,

$$\ln p_{\text{clm}}(\tau|c) = \ln \frac{p(\tau)p(c|\tau)}{p(c)} \propto \ln p_{\text{lm}}(\tau) + \ln p_{\text{cls}}(c|\tau) . \tag{2}$$

Similar to CCLM, a target code $c'$ is fed to $p_{\text{cls}}(c'|\hat{\tau})$ during inference. The differences is that $c$ is given as a label of $p_{\text{cls}}(\cdot|\tau)$ rather than a conditional variable of $p_{\text{lm}}(x_t|x_{<t}, \cdot)$. The parameters of $p_{\text{lm}}(\tau)$ and $p_{\text{cls}}(c|\tau)$ are separately pre-trained (i.e., $\theta_{\text{lm}}$ is not updated on $c$), and the decoding process is controlled online by summing up the log-likelihoods of on-the-fly sequences $\hat{\tau}$ and target codes $c'$ (i.e., sampling $\hat{\tau}$ that maximizes $\ln p_{\text{lm}}(\hat{\tau}) + \ln p_{\text{cls}}(c'|\hat{\tau})$).

### 2.2 REINFORCED LANGUAGE MODEL

Reinforced Language Models (RLM) is somewhere in the middle of CCLM and BCLM. Analogous to BCLM, the RLM separates the control part as a reward model $R(\tau) := p_{\text{cls}}(c|\tau)$, but like CCLM, $\theta_{\text{lm}}$ is updated on $c$ through $R(\tau)$,

$$\ln p_{\text{clm}}(\tau|c) \approx \ln p_{\text{lm}}(\tau|\theta_{\text{clm}}) \qquad \text{where} \qquad \theta_{\text{clm}} = \arg\max_{\theta_{\text{lm}}} \mathop{\mathbb{E}}_{\tau \sim p_{\text{lm}}(\hat{\tau}|\theta_{\text{lm}})} [R(\tau)] . \tag{3}$$

In general, RLM studies (Stiennon et al., 2020; Korbak et al., 2022; Ouyang et al., 2022; Bai et al., 2022) focuses on maximizing the objective function defined as:

$$\arg\max_{\theta} \mathop{\mathbb{E}}_{\tau \sim \pi_{\theta}^{\text{RL}}} [R(\tau)] - \text{KL}[\pi_{\theta}^{\text{RL}}(\tau) || \pi_{\bar{\theta}}^{\text{SL}}(\tau)] \iff \arg\min_{\theta} \text{KL} \left[ \pi_{\theta}^{\text{RL}}(\tau) || \pi_{\bar{\theta}}^{\text{SL}}(\tau) e^{R(\tau)} \right] \tag{4}$$

where $\pi_{\bar{\theta}}^{\text{SL}}$ is a behavior model pre-trained on a supervision dataset, $R(\cdot)$ is a reward function, and $\pi_{\theta}^{\text{RL}}$ is a target model optimized for $\pi_{\bar{\theta}}^{\text{SL}}$ and $R(\cdot)$ simultaneously. This suggests that RLM is not only a bi-objective problem, but also an off-policy RL problem where the behavior model $\pi_{\bar{\theta}}^{\text{SL}}$ determines the sampling distribution of sentences $\tau \sim \pi_{\theta}^{\text{RL}}$. Note that $R(\cdot)$ is commonly defined as a reward model $R_{\phi}$ parameterized by $\phi$, i.e., a pre-trained classifier that predicts how likely the given sentence $\tau$ contains the code of intended attributes.

## 3 PROBLEM STATEMENT

### 3.1 OPTIMIZING RLM AS BI-OBJECTIVES PROBLEM

Given an off-policy RL can be viewed as a probabilistic inference (Kappen et al., 2012; Rawlik et al., 2012; Levine, 2018), Eq (4) indicates that RLM can be addressed by the probabilistic inference framework. This framework allows us an approximate inference that estimates the target trajectory

$\tau \sim \pi \in \mathcal{T}_\pi$ under the behavior trajectory $\tau \sim \beta \in \mathcal{T}_\beta$, by minimizing Kullback-Leibler Divergence (KLD):

$$\text{KL}\left[\pi(\tau)\big|\big|\beta(\tau)e^{R(\tau)}\right] = \sum_\tau \pi(\tau)\ln\frac{\pi(\tau)}{\beta(\tau)e^{R(\tau)}} \ , \tag{5}$$

where $\beta(\tau)$ is a behavior policy and $\pi(\tau)$ is a target policy. Treating RLM as a probabilistic inference implies that a control variable is defined as an entire trajectory $\tau$, rather than an action $x_t$ as is in traditional RL frameworks. Consequently, we can optimize RLM by minimizing Eq (5), and an optimal solution $\pi^*(\tau)$ is obtained as a composite of the reward objective $R(\cdot)$ and the (behavior policy's) likelihood objective $\beta(\cdot)$:

$$\pi^*(\tau) = \frac{\beta(\tau)e^{R(\tau)}}{\sum_\tau \beta(\tau)e^{R(\tau)}} = \beta(\tau)e^{R(\tau)} \quad \left(\because \sum_\tau \beta(\tau)e^{R(\tau)} = 1\right), \tag{6}$$

which confirms that optimizing RLMs is the bi-objectives problem. Note that $\sum_\tau \beta(\tau)e^{R(\tau)} = 1$ is a necessary condition for Eq (5) to have a minimum value.

## 3.2 PARETO OPTIMIZATION PROBLEM

There are two cases of the bi-objectives problem: when the two objectives are in conflict or not. The latter case is referred to as a Pareto optimization problem (Ngatchou et al., 2005; Kyriakis & Deshmukh, 2022; Lin et al., 2019; 2022) that entails the following concepts:

**Definition 3.1** (Pareto Dominance). *Assume arbitrary policies $\pi^a, \pi^b \in \Theta_\pi$. $\pi^a$ is said to dominate $\pi^b$, denoted as $\pi^b \prec \pi^a$, if and only if $\mathbb{E}_{\tau\sim\pi^b}[R(\tau)] \leq \mathbb{E}_{\tau\sim\pi^a}[R(\tau)]$ and $\mathbb{E}_{\tau\sim\pi^b}[\beta(\tau)] \leq \mathbb{E}_{\tau\sim\pi^a}[\beta(\tau)]$ for all $\tau$.*

**Definition 3.2** (Pareto Improvement). *If $\pi^b \prec \pi^a$, the move from $\pi^b$ to $\pi^a$ is a Pareto improvement. Let $\hat{\pi}, \pi^* \in \Theta_\pi$ be non-optimal and optimal policies, respectively. A Pareto improvement always occurs for $\hat{\pi}$.*

**Definition 3.3** (Pareto Optimality). *A policy $\pi^* \in \Theta_\pi$ is Pareto optimal if there is no $\hat{\pi} \in \Theta_\pi$ such that $\pi^* \prec \hat{\pi}$, and a trajectory $\tau^* \in \mathcal{T}_{\pi^*}$ is said to be a Pareto optimal point.*

**Definition 3.4** (Pareto Set / Frontier). *A Pareto set is a set of Pareto optimal points, and its image in the objective space is the Pareto frontier.*

Simply put, the target policy $\pi(\tau)$ is called Pareto optimal when policy improvement is no longer possible. Note that an optimal policy $\pi^*(\tau)$ is the Pareto solution, a trajectory sampled from it is the Pareto optimal point $\tau^* \sim \pi^*$, and a line connecting all optimal points is called the Pareto frontier. In Section 4, we show that optimizing an RLM is the Pareto optimization problem where the reward objective $R(\cdot)$ and the likelihood objective $\beta(\cdot)$ are in a trade-off.

## 3.3 TERMS & NOTATIONS

Given that RLMs integrate both RL and LM contexts, some readers may find the context-specific terminology confusing. Therefore, in this paper, we aim to use consistent terminology across these contexts to eliminate potential confusion. In the subsequent paragraph, we define some interchangeable terms and notations used in our study

In the RL context, $\tau$ is a trajectory consisting of total $T$ actions, where each $t$-th action $a_t$ is sampled from the behavior policy $\beta(\tau)$. In the LM context, $\tau$ is a text sequence $x = [x_1, \cdots, x_T]$ consisting of total $T$ words sampled from the behavior LM $\beta_{\bar{\theta}}(\tau)$. Actions (words) are tokenized by zero or natural numbers, so the trajectory (sequence) space $\mathcal{T}$ is defined over non-negative integer space, $\tau \in \mathcal{T} \subseteq \mathbb{Z}_0$. From RLM perspectives, the two contexts have the same training goal: *"to optimize the target policy $\pi(\tau)$ or target LM $\pi_\theta(\tau)$ w.r.t. the reward objective $R(\tau)$, but adhere to the likelihood objective (i.e., behavior policy or behavior LM) $\beta(\tau)$."* The parameters of the behavior and target LMs are denoted as $\bar{\theta}$ and $\theta$, respectively. $\bar{\theta}$ is a pre-defined (pre-trained) fixed parameter and $\theta$ is a parameter that is initialized to $\bar{\theta}$ and fine-tuned during a training process. In this paper, we use the terms "policy" and "LM" interchangeably. Therefore, let us focus on whether a given policy (or LM) is either behavior or target one. Finally, we refer to $\beta(\tau)$ as the behavior policy or likelihood objective. The former name comes from the off-policy RL perspective, while the latter comes from

the Pareto optimization perspective. Note that $\pi(\tau)$ and $\beta(\tau)$ are the probability density functions, i.e., $\pi(\tau), \beta(\tau) \in [0, 1]$, $\sum_\tau \pi(\tau) = \sum_\tau \beta(\tau) = 1$. Similarly, $R(\tau) \in [0, 1]$ was considered to have values between 0 and 1.

## 4    THEORETICAL ANALYSIS, METHODOLOGY, AND VALIDATION

In this section, we prove that RLM can theoretically be considered a Pareto optimization problem. First, we show that reward is upper-bounded (see Theorem 4.1), and that the Pareto optimality is achieved by the reward upper bound such that the reward objective $R(\tau)$ is negatively logarithmic to the likelihood objective $\beta(\tau)$ (see Theorem 4.2). Furthermore, we present that the negation of the reward upper bound yields a Pareto improvement condition (see Theorem 4.3) and propose a simple yet powerful method, called Reward Dropout, that guarantees to improve the bi-objective optimization of RLM. Finally, we empirically verify the theoretical results and validate whether Reward Dropout is effective.

### 4.1    REWARD UPPER BOUND, PARETO OPTIMALITY & IMPROVEMENT

The essence of Pareto optimality lies in that both objectives have a trade-off at the optimal points, or in other words, both objectives, $R(\tau)$ and $\beta(\tau)$, must be better off simultaneously up to the optimal points. This implies one objective can be improved only up to "a certain level" without sacrificing the other; here, that level represents "an optimal state." Accordingly, if the optimal state is specified, the reward objective $R(\tau)$ should be upper-bounded by the likelihood objective $\beta(\tau)$. In this regard, we present Theorem 4.1 that provides a Reward Upper BOund (RUBO). RUBO provides us with an interesting intuition: *"the larger the KL divergence between $\pi(\tau)$ and $\beta(\tau)$, the higher the expected reward $\mathbb{E}_{\tau \sim \pi}[R(\tau)]$."* In other words, reward maximization requires $\pi$ to deviate as far away from $\beta$ as possible. The proof is given in Appendix B.1.

**Theorem 4.1** (Reward Upper Bound). *If $\sum_\tau \pi(\tau) = 1$ and $\pi(\tau) = \beta(\tau)e^{R(\tau)}$ hold, then $\mathbb{E}_{\tau \sim \pi}[R(\tau)] \leq KL\left[\pi(\tau) \middle\| \beta(\tau)\right]$ holds.*

*Proof sketch.* Show that Eq (5), i.e., $KL\left[\pi(\tau) \| \beta(\tau)e^{R(\tau)}\right]$, is non-negative and yields an inequality. Then, we can rewrite the inequality such that the reward is upper-bounded. ☐

According to Definitions 3.1 through 3.4, Pareto optimality requires that the likelihood $\beta(\tau)$ and reward $R(\tau)$ objectives should be negatively related for all the optimal points $\tau^* \sim \pi^*$, and the Pareto frontier should be drawn as a rightward sloping line accordingly. That is, the optimal policy $\pi^*(\tau)$ must yield a result that the two objectives are negatively related. In this regard, we present Theorem 4.2. Theorem 4.2 states that as long as RUBO holds, $\beta(\tau)$ and $R(\tau)$ have a negative logarithmic relationship for all optimal solutions, i.e., $\forall \tau^* \sim \pi^*$ where $\pi^*$ is given by Eq (6), i.e., $\pi^*(\tau) = \beta(\tau)e^{R(\tau)}$. This clarifies that optimizing RLM is a Pareto optimization problem. The proof is given in Appendix B.2.

**Theorem 4.2** (Pareto Optimality). *If $\mathbb{E}_{\tau \sim \pi}[R(\tau)] \leq KL\left[\pi(\tau) \middle\| \beta(\tau)\right]$ holds, then $\forall \tau^* \sim \pi^*$, $R(\tau) = -\ln \beta(\tau)$ holds.*

*Proof sketch.* Show that when the expected reward is maximized, i.e., $\max \mathbb{E}_{\tau \sim \pi}[R(\tau)]$, the inequality by reward upper bound, i.e., $\mathbb{E}_{\tau \sim \pi}[R(\tau)] \leq KL\left[\pi(\tau) \| \beta(\tau)\right]$, becomes equality, and that $R(\tau)$ and $\beta(\tau)$ are negatively related in this equality. ☐

In a Pareto optimization problem, any policy that is not Pareto optimal always has room for Pareto improvement. By Theorems 4.1 and 4.2, we confirmed that Eq (6), i.e., $\pi(\tau) = \beta(\tau)e^{R(\tau)}$, is a Pareto optimal, and thus, by Definition 3.2, $\pi(\tau) \neq \beta(\tau)e^{R(\tau)}$ always results in the Pareto improvement. That is, satisfying a condition for $\pi(\tau) \neq \beta(\tau)e^{R(\tau)}$ guarantees to improve the optimization of RLM. We derive this condition through *reductio ad absurdum*, i.e., the proof by contradiction, in Theorem 4.3. Specifically, Theorem 4.3 shows that the negation of Theorem 4.1, i.e., $\mathbb{E}_{\tau \sim \pi}[R(\tau)] > KL[\pi(\tau) \| \beta(\tau)]$, leads to the negation of Eq (6), i.e., $\pi(\tau) \neq \beta(\tau)e^{R(\tau)}$, and the Pareto improvement holds accordingly. As shown by the proof in Appendix B.3, $\mathbb{E}_{\tau \sim \pi}[R(\tau)] > KL[\pi(\tau) \| \beta(\tau)]$ is equivalent to $\mathbb{E}_{\tau \sim \pi_\theta}[R(\tau)] + \mathbb{E}_{\tau \sim \pi_\theta}[\ln \beta(\tau)] > 0$, and thus updating $\theta$ to satisfy $\mathbb{E}_{\tau \sim \pi_\theta}[R(\tau)] + \mathbb{E}_{\tau \sim \pi_\theta}[\ln \beta(\tau)] > 0$ guarantees policy improvement. In short, we can better optimize $\theta$ by manipulating $R(\cdot)$ or $\beta(\cdot)$ such that $\mathbb{E}_{\tau \sim \pi}[R(\tau)] + \mathbb{E}_{\tau \sim \pi}[\ln \beta(\tau)] > 0$ is always satisfied.

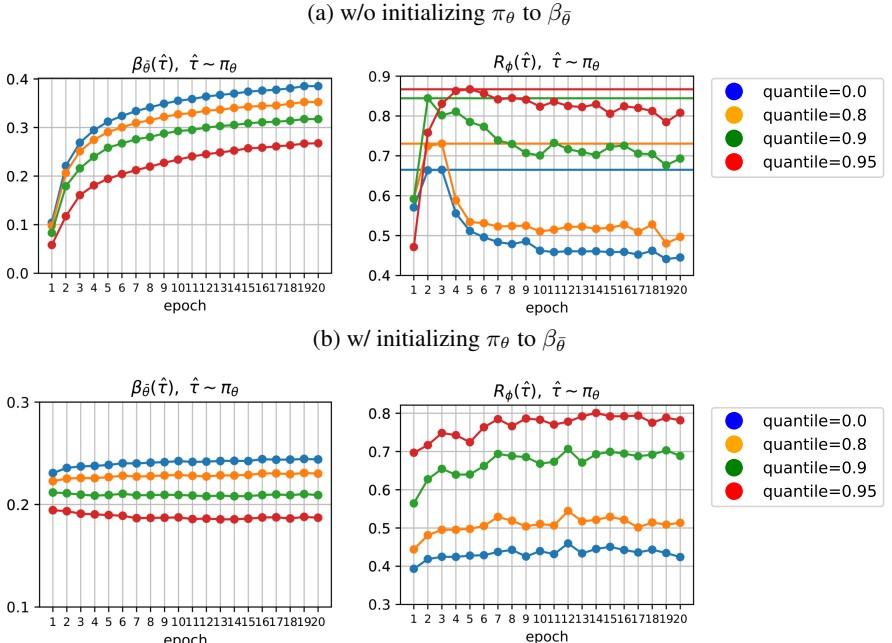

Figure 1: Presenting the bi-objectivity and trade-off in Eq (5), and illustrating the Reward Dropout effect according to different quantiles and scenarios. The red, green, orange, and blue horizontal lines represent the Reward Upper BOund (RUBO). It is worth noting that Reward Dropout works consistently in both scenarios (a) and (b).

**Theorem 4.3** (Pareto Improvement Condition). *For $\pi(\tau) \neq \beta(\tau)e^{R(\tau)}$ to hold, $\mathbb{E}_{\tau \sim \pi}[R(\tau)] > KL\left[\pi(\tau)\middle\|\beta(\tau)\right]$, or equivalently Eq (7), must hold.*

$$\mathbb{E}_{\tau \sim \pi}[R(\tau)] + \mathbb{E}_{\tau \sim \pi}[\ln \beta(\tau)] > 0 \tag{7}$$

*Proof sketch.* Show that the contraposition of RUBO, i.e., $\mathbb{E}_{\tau \sim \pi}[R(\tau)] > KL\left[\pi(\tau)\middle\|\beta(\tau)\right]$, yields the Pareto improvement, i.e., $\pi(\tau) \neq \beta(\tau)e^{R(\tau)}$. For this, we can use *"the proof by contradiction"* method: $p \to q \;\Rightarrow\; \neg q \to \neg p$. □

### 4.2 REWARD DROPOUT

According to Theorem 4.3, the target policy is guaranteed to improve both $R(\tau)$ and $\beta(\tau)$ simultaneously as long as Eq (7) holds. The message behind it is simple: *"all you need is to manipulate either or both $R(\cdot)$ and $\beta(\cdot)$ so that $\mathbb{E}_{\tau \sim \pi}[R(\tau)] + \mathbb{E}_{\tau \sim \pi}[\ln \beta(\tau)] > 0$ is achieved for all $\tau$."* However, in an off-policy RL context, $\beta(\tau)$ is either a pre-defined policy or a pre-trained LM, whose distribution or parameter should be fixed. Accordingly, it is only $R(\tau)$ that can be manipulated, and *"we need to manipulate $R(\tau)$ such that only a few high rewards are considered."* The reason why "only a few high rewards" should be considered is that $\mathbb{E}_{\tau \sim \pi}[R(\tau)]$ refers to the average reward $\tilde{r}$, and the average is sensitive to bias caused by outliers. Extremely saying, a single high reward is more effective at satisfying Eq (7) than many average rewards. As a practical implementation of reward manipulation, we proposed Reward Dropout, a technique that leaves only a few high rewards and sets the rest to zero. Specifically, it sorts rewards in ascending order, divides them into equal intervals, and then sets rewards that fall below a certain quartile to zero. Reward Dropout is an example of an RL technique that leverages quantized reward intervals (Dabney et al., 2018; Lu et al., 2022), and is therefore applicable to any models or algorithms that deal with RL problems.

### 4.3 BI-OBJECTIVITY, TRADE-OFF, AND THE EFFECT OF REWARD DROPOUT

In this section, we present empirical evidence that optimizing RLM is a Pareto optimization problem where *"the two conflicting objectives are optimized simultaneously."* Also, we show that satisfying Eq (7), or Reward Dropout, is indeed effective for improving the optimization of RLM. Lastly, we experiment with how the initialization of the target policy affects the optimization performance.

To evaluate the performance, we visualized a trend of likelihood and reward objectives. Both objectives were represented by behavior model $\beta_{\bar{\theta}}$ and reward model $R_\phi$, respectively. For the behavior model we used a benchmark LLM (e.g., OpenAI GPT-2) without fine-tuning, while for the reward model, we implemented a Transformer-based classifier by ourselves and pre-trained it on the relevant dataset according to control attributes (e.g., sentiment, topic, etc.). In this experiment, we used AG_News (Zhang et al., 2015) dataset, and the reward model was trained to predict how likely the given sentence belongs to a sports topic. Figure 1 shows the result in two different scenarios: (a) $\pi_\theta$ was initialized with random parameters, and (b) $\pi_\theta$ was initialized with behavior parameters $\bar{\theta}$.

Figure 1a illustrates a result of scenario (a) and demonstrates that both objectives are maximized simultaneously (i.e., bi-objectives optimization) until the RUBO is touched, after which we can observe that $R(\cdot)$ continues to fall while $\beta(\cdot)$ continues to rise (i.e., trade-offs between $\beta(\cdot)$ and $R(\cdot)$). This supports Theorems 4.1 and 4.2 that (1) the reward is upper-bounded (2) $\beta(\cdot)$ and $R(\cdot)$ are log-negatively related, and provides evidence that RLM is a Pareto optimization problem. On the other hand, Figure 1b illustrates a result of scenario (b). In general, RLM researchers prefer to follow this scenario (so do we) because if we initialize the target policy $\pi_\theta$ to the pre-trained LLM $\beta_{\bar{\theta}}$, we do not need to optimize our model for the likelihood objective but only for the reward objective; the large parameter space of LLM is robust enough to withstand parameter degeneracy due to reward optimization. For training practicality and stability, all experiments in this paper were designed to follow scenario (b).

The key takeaway from Figure 1 is that Reward Dropout drives performance improvement in both scenarios (a) and (b). It is also noteworthy that as the quantile increases, i.e., as the model learns rewards that are biased toward the top few outliers, performance improves more significantly. This is exactly what we intended for Reward Dropout, as described in Section 4.2. Most importantly, the sum of the likelihood and reward values always increases with Reward Dropout, which is evidence that Reward Dropout is definitely achieving Pareto Improvement.

## 5 BENCHMARK EXPERIMENTS

In this section, we evaluate the performance of Reward Dropout on five benchmark datasets and test whether the effect of Reward Dropout maintains regardless of the capacity of behavior LMs.

**Datasets** To validate the effectiveness of Reward Dropout, we conducted performance experiments on five RLM benchmark datasets, aiming to control the generation of text with specific attributes. Each dataset covers different attributes of sentences including sentiment (negative, positive), politeness (polite, non-polite), toxicity (toxic, non-toxic), emotion (anger, disgust, fear, happiness, sadness, surprise), and topic (world, sports, business, sci/tech). For the sentiment, toxicity, emotion, and topic datasets, we collected publicly accessible sources such as Yelp (Zhang et al., 2015), Jigsaw (Dataset, 2017), DailyDialog (Li et al., 2017a), and AG_News (Zhang et al., 2015), respectively. The politeness dataset was downloaded from the GitHub repository released by Madaan et al. (2020).[1]

**Models & Algorithms.** To build the behavior LM $\beta_{\bar{\theta}}$, we used OpenAI GPT-2 (Radford et al., 2019), the pre-trained LLM released by HuggingFace transformers library.[2] The target LM $\pi_\theta$ were initialized to the parameters of $\beta_{\bar{\theta}}$, and the target parameters $\theta$ were updated by fine-tuning them *w.r.t* the rewards predicted by a pre-trained reward model $R_\phi$. For update algorithms, we utilized three policy-based RL algorithms: deterministic policy gradient (DPG) (Silver et al., 2014), stochastic policy gradient (SPG) (Williams, 1992; Sutton et al., 1999), and top-k policy gradient (KPG). They were all implemented in an off-policy gradient fashion (Degris et al., 2012; Liu et al., 2020b) (see Algorithm 1). In the LM context, we can implement DPG and SPG with greedy decoding and stochastic decoding, respectively. Similarly, the KPG was implemented based on top-k decoding strategy, expecting an intermediate performance between DPG and SPG. See Appendix C.1 for more information on how to implement DPG, SPG, and KPG in the LM context.

**Random Dropout and Dropout Rate** To clarify what we are dropping out affects the performance of Reward Dropout, we introduced a random Reward Dropout inspired by Srivastava et al. (2014). Random Dropout randomly sets some rewards to zero according to the dropout rate. In

---

[1] https://github.com/tag-and-generate/politeness-dataset
[2] https://huggingface.co/gpt2

| Decoding (Policy Gradient) | Reward Dropout | $\gamma$ | Dataset | | | | |
|---|---|---|---|---|---|---|---|
| | | | sentiment | politeness | toxicity | emotion | topic |
| *greedy (DPG)* | *no dropout* | – | 0.506 | 0.602 | 0.505 | 0.023 | 0.277 |
| | *random* | 0.80 | 0.512 | 0.641 | 0.513 | 0.024 | 0.298 |
| | | 0.90 | 0.513 | 0.652 | 0.513 | 0.026 | 0.302 |
| | | 0.95 | 0.514 | 0.663 | 0.512 | 0.024 | 0.304 |
| | *quantile* | 0.80 | 0.735 | 0.715 | 0.521 | 0.049 | 0.496 |
| | | 0.90 | 0.780 | 0.834 | 0.529 | 0.062 | 0.609 |
| | | 0.95 | 0.778 | 0.883 | 0.562 | 0.067 | 0.688 |
| *stochastic (SPG)* | *no dropout* | – | 0.660 | 0.896 | 0.706 | 0.103 | 0.489 |
| | *random* | 0.80 | 0.652 | 0.891 | 0.719 | 0.096 | 0.500 |
| | | 0.90 | 0.662 | 0.894 | 0.700 | 0.110 | 0.494 |
| | | 0.95 | 0.654 | 0.903 | 0.707 | 0.089 | 0.492 |
| | *quantile* | 0.80 | 0.821 | 0.933 | 0.741 | 0.141 | 0.607 |
| | | 0.90 | 0.852 | 0.950 | 0.759 | 0.166 | 0.712 |
| | | 0.95 | **0.854** | **0.971** | **0.785** | **0.192** | **0.777** |
| *top-k (KPG)* | *no dropout* | – | 0.677 | 0.864 | 0.671 | 0.089 | 0.500 |
| | *random* | 0.80 | 0.669 | 0.877 | 0.665 | 0.089 | 0.497 |
| | | 0.90 | 0.672 | 0.876 | 0.704 | 0.093 | 0.493 |
| | | 0.95 | 0.668 | 0.875 | 0.687 | 0.088 | 0.493 |
| | *quantile* | 0.80 | 0.833 | 0.892 | 0.703 | 0.111 | 0.617 |
| | | 0.90 | **0.861** | 0.930 | 0.722 | 0.129 | 0.711 |
| | | 0.95 | 0.858 | 0.963 | 0.741 | 0.145 | 0.770 |

Table 1: The numbers in the table denote the average rewards at the end of training. The underlined, red-colored, and **bolded** numbers represent the highest performance cases across the dropout, decoding, and dataset options, respectively. Note that $\gamma$ is the dropout rate.

| Dataset | Control attribute | Generated text |
|---|---|---|
| *sentiment* | *negative* | The chicken-crap, which is the worst thing I've ever seen. |
| | *positive* | The chicken is so delicious, it's a big one. |
| *topic* | *world* | The issue focused on the fact that Iran is not a state of war, and it has been unable to defend its people. |
| | *sci/tech* | The issue focused on the development of a new system for computing and networking is that it takes more than two seconds to develop. |

Table 2: Above texts were generated by the target LM trained with stochastic decoding and quantile dropout ($\gamma = 0.95$). The underlined phrase refers to a given prefix, and the red-colored words highlight controlled parts. More examples are provided in Appendix F

addition, to evaluate how performance changes with the dropout rate, we introduced $\gamma$ as a hyperparameter that denotes the percentage of zero rewards per training batch. Three dropout rates $\gamma \in \{0.80, 0.90, 0.95\}$ were tested.

**Implementation Details**   In order for the behavior LM to generate a sentence, we need to provide the behavior LM with an initial state to start the generation process. To do this, we provided the behavior LM with a prefix that is an incomplete sentence as an initial state. Refer to Appendix C for implementation details (e.g., pseudo algorithm, hyperparameters, initialization setting, etc.).

**Evaluation**   The performance of Reward Dropout was evaluated in three ways. First, we compared the average rewards of target LM at the end of training (see Table 1). This summarizes the expected

rewards achieved at the Pareto optimal state. Second, we visualized the reward growth of the target LM over training epochs. This describes the Pareto improvement effect (see Appendix E) driven by Reward Dropouts. Third, the controlled texts were evaluated by humans to ensure if they are reliable. For fairness, we grouped the evaluators to represent as different genders and races as possible (see Appendix G). Lastly, we tested if the larger behavior LM, the weaker the effect of Reward Dropout. The first and second evaluations were conducted with different dropout, decoding, and hyperparameter settings, while the third and last evaluation was conducted with the best setting.

## 6 RESULTS

Table 1 shows the results for our first evaluation: Reward Dropout improves the control performance for all decodings and datasets. In particular, it is likely that the higher $\gamma$ (i.e., the more dropout), the better performance. Also, we can observe that the quantile dropout is much more effective than the random dropout, which is evidence that reward manipulation leads to Pareto improvement. Table 2 presents some examples of generated text that was controlled to have a specific sentiment or topic. These examples show that stochastic decoding with quantile dropout successfully controls the target LM to generate text as intended.

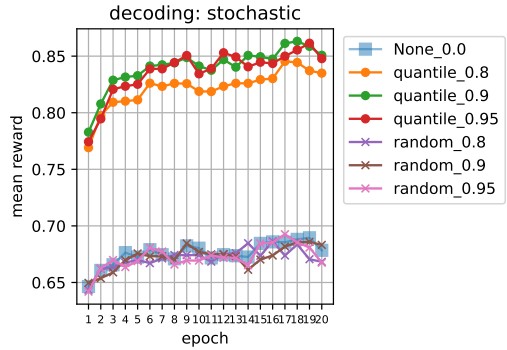

Figure 2: Sentiment Control - Negative

Figure 2 is a case result related to the second evaluation, showing that the average reward of the target LM increases throughout training. This suggests that quantile dropout is undoubtedly effective. We provide the full results in Appendix E due to page limit. To summarize the full results, 1) greedy decoding is the worst decoding strategy while stochastic decoding is the best one, 2) random dropout improves control performance better than no dropout at least with greedy decoding, and 3) there is no outstanding trend of reward growth in the emotion dataset. The last point is probably due to the unbalanced labels and lack of samples in the emotion dataset (see Appendix D), which also explains the small figures at emotion column in Table 1.

The third evaluation was conducted through a survey. We prepared 55 items designed to ask three types of questions: 1) distinguish between real and generated text, 2) select the more human-like text, and 3) label appropriate control attributes (i.e., control codes) to the generated text. Due to the page limit, we provide the survey form and results in Appendix G. The survey results show that respondents confused real texts and generated texts, and believed that generated text is more human-like. At the same time, it showed that the control performance met humans' reliability standards. In conclusion, training a target LM with stochastic decoding and quantile dropout can produce reliable texts with human-level control.

Figure 3 illustrates how the parameter size of LLMs affects the performance of Reward Dropout. We compared four models in total, with the models and parameter sizes as follows:

- OpenAI GPT2 (117 million parameters)
- Meta OPT (350 million (Zhang et al., 2022b))
- Meta XGLM (564 million (Lin et al., 2021))
- MIT GPT2 (774 million)

This shows that the effect of Reward Dropouts is always valid regardless of the parameter size. Reward Dropout always outperformed the non-dropout case (q=0.0). The effect between dropout rates was almost consistent. In 7 out of the 8 cases, higher dropout rates led to better performance. The only exception was XGLM with the Topic - Sci/Tech dataset, where the effect between dropout rates was reversed. Also, we can see that the parameter size of the LLM has a positive impact on the RLM optimization, in particular, the larger the model capacity, the higher the average reward. The exception was once again XGLM, and one possible explanation for this is that XGLM was pre-trained on a multilingual translation dataset. Given that our experiment was conducted with English sentences only, the large number of languages the model had to learn with its limited parameter

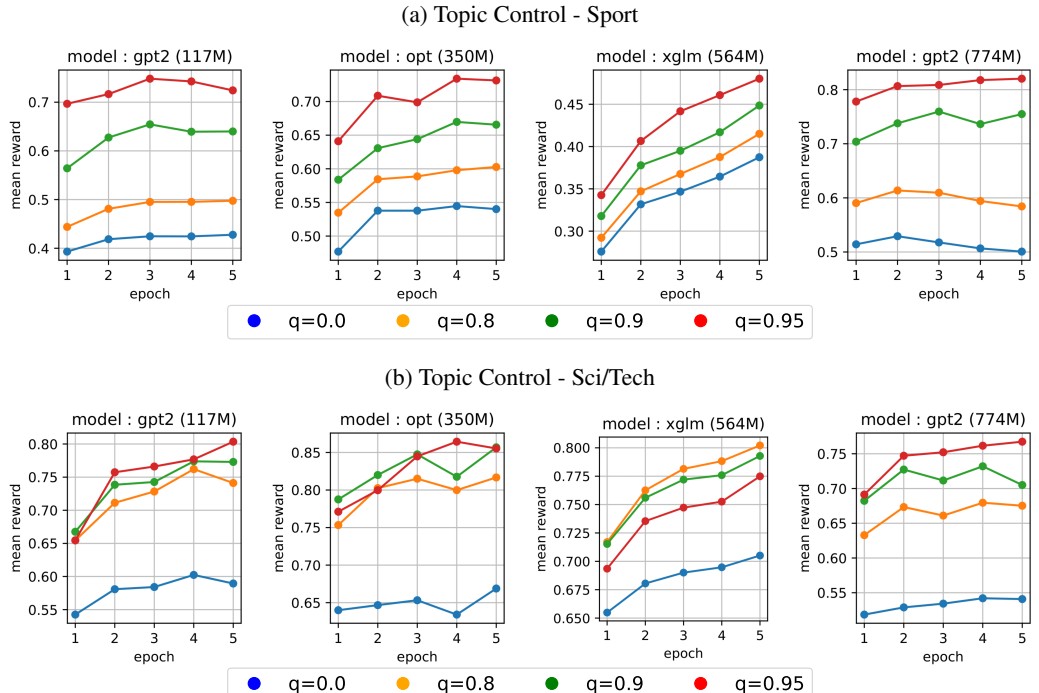

Figure 3: Comparing the effect of Reward Dropout using different LLMs

capacity resulted in poorer quality sentence generation, which may have contributed to the poor performance. We believe further analysis is needed in this regard.

## 7 LIMITATIONS & CONCLUDING REMARKS

In this study, we (1) laid a theoretical foundation for RLMs from a bi-objective perspective, (2) presented theoretical and empirical evidence that optimizing RLM is indeed a Pareto optimization problem, (3) proposed a simple yet powerful method named Reward Dropout that guarantees to improve the bi-objective optimization of RLMs, and (4) demonstrated the effect of Reward Dropout is consistent across five benchmark datasets and four benchmark LLMs. Not only is Reward Dropout theoretically sound and easy to implement, but its effects were validated powerful as expected.

Meanwhile, among the different approaches to developing controllable language models (CLMs), Reward Dropout is only applicable to the RLM class. RLMs have the obvious limitation of high training-time complexity and the need to build and train separate models for each control attribute. However, despite the high training-time complexity of RLM, there is a large body of RLM literature. This is because, under the RL framework, controllability is always guaranteed through the policy improvement theorem. This implies that the decision to use RLMs is a matter of choosing between training efficiency and guaranteed controllability.

At this point, we believe that the value of Reward Dropout comes into play again, because it is a technique that guarantees to improve the optimization of the RLM, which in turn improves training efficiency. Beyond its training efficiency, Reward Dropout can be applied to any model, algorithm, or neural network structure that deals with problems of reward maximization, or problems that can be formalized as a reinforcement learning framework. Therefore, we believe that Reward Dropout can make a significant contribution to the field of artificial intelligence research and development. For reproducibility, we release our code at https://github.com/anonymous-user01.

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

## A   DERIVATIONS

### A.1   DERIVATION OF EQUATION (6)

From a probabilistic inference perspective, the off-policy RL can be addressed as a problem of minimizing the following Kullback-Leibler Divergence (KLD) (Kappen et al., 2012; Levine, 2018):

$$
\text{KL}\left[\pi(\tau)||\beta(\tau)e^{R(\tau)}\right] = \sum_\tau \pi(\tau)\ln\frac{\pi(\tau)}{\beta(\tau)e^{R(\tau)}} = -\mathop{\mathbb{E}}_{\tau\sim\pi}[R(\tau)+\ln\beta(\tau)] - \mathcal{H}\left[\pi\right] . \tag{8}
$$

Here, $R(\tau)$ is the reward function defined by the control objective, which is set to increase the reward signal exponentially. From RL perspectives, minimizing Eq (8) can be converted into a maximization problem,

$$
\arg\max_\pi \mathop{\mathbb{E}}_{\tau\sim\pi}[R(\tau)+\ln\beta(\tau)] + \mathcal{H}\left[\pi\right] \qquad s.t. \quad \sum_\tau \pi(\tau) = 1 ,
$$

where $\sum\pi(\tau) = 1$ is the constraint that the total probability of the sampled trajectories must be 1, and, by the Lagrangian method, the objective function is written as

$$
\mathcal{L}(\pi) = \mathop{\mathbb{E}}_{\tau\sim\pi}[R(\tau)+\ln\beta(\tau)] + \mathcal{H}\left[\pi\right] + \lambda\left(\sum_\tau \pi(\tau) - 1\right) . \tag{9}
$$

By optimizing Eq (9), the optimal policy $\pi^*$ is given such that both the reward and the likelihood of behavior policy are maximized w.r.t. a sampled trajectory from target policy $\tau\sim\pi(\tau)$,

$$
\pi^*(\tau) = \beta(\tau)e^{R(\tau)} \times e^{-(1-\lambda)} = \frac{\beta(\tau)e^{R(\tau)}}{e^{1-\lambda}} = \frac{\beta(\tau)e^{R(\tau)}}{\sum_\tau\beta(\tau)e^{R(\tau)}} . \tag{10}
$$

Note that $e^{1-\lambda}$ is a normalization constant (partition function) defined by the probability condition $\sum_\tau\pi(\tau) = 1$. This implies that if the Lagrange multiplier is equal to 1, i.e., $\lambda = 1$, the optimal target policy $\pi^*(\tau)$ follows $\beta(\tau)e^{R(\tau)}$ because $\pi^*(\tau) = \beta(\tau)e^{R(\tau)}/e^0 = \beta(\tau)e^{R(\tau)}$, and the trajectory sampling $\tau\sim\pi(\tau)$ is determined by $\beta(\tau)e^{R(\tau)}$.

## B   PROOFS

### B.1   PROOF OF THEOREM 4.1

**Theorem 4.1**   *If $\sum_\tau\pi(\tau) = 1$ and $\pi(\tau) = \beta(\tau)e^{R(\tau)}$ hold, then $\mathbb{E}_{\tau\sim\pi}[R(\tau)] \leq KL\left[\pi(\tau)||\beta(\tau)\right]$ holds.*

*Proof.* Given $\sum_\tau\pi(\tau) = 1$ and $\pi(\tau) = \beta(\tau)e^{R(\tau)}$, it is obvious that $\sum_\tau\pi(\tau) = \sum_\tau\beta(\tau)e^{R(\tau)} = 1$ holds. Then, we can obtain the non-negativity of KL $\left[\pi(\tau)||\beta(\tau)e^{R(\tau)}\right]$.

$$
\begin{aligned}
\text{KL}\left[\pi(\tau)||\beta(\tau)e^{R(\tau)}\right] &= \sum_\tau \pi(\tau)\ln\frac{\pi(\tau)}{\beta(\tau)e^{R(\tau)}} = -\sum_\tau \pi(\tau)\ln\frac{\beta(\tau)e^{R(\tau)}}{\pi(\tau)} \\
&\geq -\ln\sum_\tau \pi(\tau)\frac{\beta(\tau)e^{R(\tau)}}{\pi(\tau)} \qquad (\because \text{ Jensen Inequality}) \\
&= -\ln\sum_\tau \beta(\tau)e^{R(\tau)} = 0 \qquad \left(\because \sum_\tau\beta(\tau)e^{R(\tau)} = 1\right)
\end{aligned}
$$

As a result, KL $\left[\pi(\tau)||\beta(\tau)e^{R(\tau)}\right] \geq 0$ holds, which leads to $\mathbb{E}_{\tau\sim\pi}[R(\tau)] \leq$ KL $[\pi(\tau)||\beta(\tau)]$, or equivalently, $\mathbb{E}_{\tau\sim\pi}[\ln\beta(\tau)] \leq \mathbb{E}_{\tau\sim\pi}\left[\pi(\tau)||e^{R(\tau)}\right]$.   □

## B.2 PROOF OF THEOREM 4.2

**Theorem 4.2** *If $\mathbb{E}_{\tau \sim \pi}[R(\tau)] \leq KL\left[\pi(\tau)\big|\big|\beta(\tau)\right]$ holds, then $\forall \tau^* \sim \pi^*$, $R(\tau) = -\ln \beta(\tau)$ holds.*

*Proof.* Suppose the reward upper bound is given by $\mathbb{E}_{\tau \sim \pi}[R(\tau)] \leq \text{KL}[\pi(\tau)||\beta(\tau)]$. Then, we can expand the upper bound by $-\mathbb{E}_{\tau \sim \pi}[\ln \beta(\tau)]$ as below.

$$\mathop{\mathbb{E}}_{\tau \sim \pi}[R(\tau)] \leq \text{KL}[\pi(\tau)||\beta(\tau)] = -\mathop{\mathbb{E}}_{\tau \sim \pi}[\ln \beta(\tau)] - \underbrace{\mathcal{H}[\pi]}_{\geq 0} \leq -\mathop{\mathbb{E}}_{\tau \sim \pi}[\ln \beta(\tau)]$$

This implies $\text{KL}[\pi(\tau)||\beta(\tau)] = -\mathbb{E}_{\tau \sim \pi}[\ln \beta(\tau)]$ holds iff $\mathcal{H}[\pi] = 0$. At the optimal points $\tau^* \sim \pi^*$, trivially $\mathcal{H}[\pi]$ goes to zero because the target policy converges to a specific distribution without uncertainty, which yields the maximal expected reward, i.e., $\mathbb{E}_{\tau^* \sim \pi^*}[R(\tau)] = \max \mathbb{E}_{\tau \sim \pi}[R(\tau)]$. As a result, the following identity is obtained:

$$\mathop{\mathbb{E}}_{\tau^* \sim \pi^*}[R(\tau)] = -\mathop{\mathbb{E}}_{\tau^* \sim \pi^*}[\ln \beta(\tau)], \tag{11}$$

implying $\forall \tau^* \sim \pi^*$, $R(\tau) = -\ln \beta(\tau)$ holds. $\qquad\square$

*Proof.* Here we provide another way of proving Theorem 4.2. This is more intuitive and simpler way. Let us take a partial derivative of $\text{KL}[\pi(\tau)||\beta(\tau)e^\tau]$ w.r.t $\pi(\tau)$ and set it to zero.

$$\frac{\partial \text{KL}\left[\pi(\tau)||\beta(\tau)e^{R(\tau)}\right]}{\partial \pi(\tau)} = \ln \pi(\tau) + 1 - \ln \beta(\tau) - R(\tau) \overset{\text{set}}{=} 0. \tag{12}$$

The above equation describes an implicit function of $\pi(\tau)$, $\beta(\tau)$, and $R(\tau)$, which implies the optimal state of target policy, $\pi^*(\tau)$. Now, we can arrange this in the form of an explicit function whose $R(\tau)$ is the dependent variable and the others are independent variables. Then, the result shows that $R(\tau)$ is negative logarithmic to $\beta(\tau)$,

$$R(\tau) = \ln \pi(\tau) + 1 - \ln \beta(\tau) \implies R(\tau) = -\ln \beta(\tau), \tag{13}$$

where $\ln \pi(\tau)$ and $+1$ can be ignored because they are irrelevant to interpreting the relationship between $R(\tau)$ and $\beta(\tau)$. We know that Eq (13) was derived from Eq (12) and thus implies an optimal state of target policy by itself. In other words, Eq (13) describes the condition under which Pareto optimality is achieved, and that condition is that $R(\tau)$ and $\beta(\tau)$ must be negatively related for all $\tau$. $\qquad\square$

## B.3 PROOF OF THEOREM 4.3

**Theorem 4.3** *For $\pi(\tau) \neq \beta(\tau)e^{R(\tau)}$ to hold, $\mathbb{E}_{\tau \sim \pi}[R(\tau)] > KL\left[\pi(\tau)\big|\big|\beta(\tau)\right]$, or equivalently Eq (14), must hold.*

$$\mathop{\mathbb{E}}_{\tau \sim \pi}[R(\tau)] + \mathop{\mathbb{E}}_{\tau \sim \pi}[\ln \beta(\tau)] > 0 \tag{14}$$

*Proof.* By Theorem 4.1, we know that $\pi(\tau) = \beta(\tau)e^{R(\tau)}$ is a necessary condition for the reward upper bound to hold ($p \to q$). By Definition 3.2, we also know that $\pi(\tau) \neq \beta(\tau)e^{R(\tau)}$ indicates the Pareto improvement. As a corollary, if the negation of the reward upper bound yields $\pi(\tau) \neq \beta(\tau)e^{R(\tau)}$ ($\neg q \to \neg p$), then we can assume that negation leads to the Pareto improvement.

Suppose the reward upper bound is given by $\mathbb{E}_{\tau \sim \pi}[R(\tau)] \leq \text{KL}\left[\pi(\tau)\big|\big|\beta(\tau)\right]$. We can rewrite it in the form of inequality as below:

$$0 \geq \mathop{\mathbb{E}}_{\tau \sim \pi}[R(\tau)] - \text{KL}\left[\pi(\tau)\big|\big|\beta(\tau)\right] = \mathop{\mathbb{E}}_{\tau \sim \pi}[R(\tau)] - \underbrace{\mathop{\mathbb{E}}_{\tau \sim \pi}[\ln \pi(\tau)]}_{= -\mathcal{H}[\pi] \leq 0} + \mathop{\mathbb{E}}_{\tau \sim \pi}[\ln \beta(\tau)]$$

$$\geq \mathop{\mathbb{E}}_{\tau \sim \pi}[R(\tau)] + \mathop{\mathbb{E}}_{\tau \sim \pi}[\ln \beta(\tau)] \tag{15}$$

The above inequality shows that $\mathbb{E}_{\tau \sim \pi}[R(\tau)] + \mathbb{E}_{\tau \sim \pi}[\ln \beta(\tau)]$ cannot be larger than zero as long as $\mathbb{E}_{\tau \sim \pi}[R(\tau)] \leq \text{KL}\left[\pi(\tau)\big|\big|\beta(\tau)\right]$ holds. But assume $\mathbb{E}_{\tau \sim \pi}[R(\tau)] + \mathbb{E}_{\tau \sim \pi}[\ln \beta(\tau)]$ is larger than

zero, i.e, $\mathbb{E}_{\tau \sim \pi}[R(\tau)] + \mathbb{E}_{\tau \sim \pi}[\ln \beta(\tau)] > 0$, which is equivalent to the negation of Eq (15). Then, we can see that a contradiction arises in the necessary condition $\pi(\tau) = \beta(\tau)e^{R(\tau)}$ as follows:

$$\pi(\tau) = \beta(\tau)e^{R(\tau)} \implies \ln \pi(\tau) = R(\tau) + \ln \beta(\tau) \qquad \text{(Logarithm on both sides.)}$$

$$\implies \underbrace{\mathbb{E}_{\tau \sim \pi}[\ln \pi(\tau)]}_{= -\mathcal{H}[\pi] \leq 0} = \underbrace{\mathbb{E}_{\tau \sim \pi}[R(\tau)] + \mathbb{E}_{\tau \sim \pi}[\ln \beta(\tau)]}_{\text{is assumed to be } > 0} \qquad \text{(Expectation on both sides.)}$$

$$\implies -\mathcal{H}[\pi] \neq \mathbb{E}_{\tau \sim \pi}[R(\tau)] + \mathbb{E}_{\tau \sim \pi}[\ln \beta(\tau)] \qquad \text{(Equality does not hold.)}$$

where $-\mathcal{H}[\pi] \leq 0$ and $\mathbb{E}_{\tau \sim \pi}[R(\tau)] + \mathbb{E}_{\tau \sim \pi}[\ln \beta(\tau)] > 0$ conflict. This contradiction means $\pi(\tau) \neq \beta(\tau)e^{R(\tau)}$. Since $\pi(\tau) \neq \beta(\tau)e^{R(\tau)}$ indicates the Pareto improvement, we can conclude that $\mathbb{E}_{\tau \sim \pi}[R(\tau)] + \mathbb{E}_{\tau \sim \pi}[\ln \beta(\tau)] > 0$ guarantees policy improvement. □

## C  IMPLEMENTATION DETAILS

### C.1  OFF-POLICY DPG, SPG, AND KPG IN THE LM CONTEXT

Off-policy policy gradient is an off-policy extension of the policy gradient method (Sutton & Barto, 2018; Sutton et al., 1999), and began to attract attention from Degris et al. (2012). Since then, further studies have been established to improve training efficiency (Lillicrap et al., 2015; Silver et al., 2014), correct the distribution mismatch between the behavior and target policies (Islam et al., 2019; Liu et al., 2020b), or address the sub-optimality issue of the behavior policy (Imani et al., 2018).

According to Silver et al. (2014), we can implement the off-policy DPG as below:

$$\nabla_\theta J_{\beta_{\bar{\theta}}}(\pi_\theta) \approx \sum_{s \in \mathcal{S}} \rho^{\beta_{\bar{\theta}}}(s) \nabla_\theta \pi_\theta(a|s) Q^{\pi_\theta}(s, a) ds = \mathbb{E}_{s \sim \rho^{\beta_{\bar{\theta}}}} \left[ \nabla_\theta \pi_\theta(s) \nabla_a Q^{\pi_\theta}(s, a)|_{a = \pi_\theta(s)} \right] \quad (16)$$

where $\rho^{\beta_{\bar{\theta}}}(s)$ is the state visitation history of the behavior policy $\beta_{\bar{\theta}}$, and $Q^{\pi_\theta}(s, a)$ is the state-action value function. In the LM context, we can define the $t$-th action as the $t$-th token and the $t$-th state as the partial sentence observed up to the $t$-th token. On the other hand, since a sentence has meaning as a whole, we cannot define the reward of the $t$-th token, but only the reward of the entire sentence. That is, the off-policy policy gradient cannot be established without integrating the state $s$ and action $a$ in the LM context. Therefore, we can rewrite the off-policy DPG, or Eq (16) w.r.t $\tau$ as follows:

$$\nabla_\theta J_{\beta_{\bar{\theta}}}(\pi_\theta) = \mathbb{E}_{\tau \sim \beta_{\bar{\theta}}}[\nabla_\theta \pi_\theta(\tau) R_\phi(\tau)] \qquad s.t. \quad \tau = \arg\max_\tau \beta_{\bar{\theta}}(\tau). \quad (17)$$

Note that $s$ and $a$ were integrated into $\tau$, meaning that we considered the entire trajectory $\tau$ instead of intermediate actions and states, and excluded the intervention of the target policy $\pi_\theta$ within the trajectory.[3] As a result, 1) the state-action value function $Q^{\pi_\theta}(s, a)$ was replaced by the reward model $R_\phi(\tau)$, 2) the action derivative $\nabla_a$ was removed, 3) the state visitation history of the behavior policy $s \sim \rho^{\beta_{\bar{\theta}}}$ was replaced the behavior trajectory $\tau \sim \beta_{\bar{\theta}}(\tau)$, and lastly, 4) the deterministic target policy $a = \pi_\theta(s)$ was removed, and the argmax constraint $\tau = \arg\max_\tau \beta_{\bar{\theta}}(\tau)$ was introduced instead. Given $\arg\max_\tau \beta_{\bar{\theta}}(\tau)$ represents greedy decoding, Eq (17) implies that we can implement the off-policy DPG by running the greedy decoding strategy. In other words, we can also implement the off-policy SPG and KPG simply by removing the argmax constraint from Eq (17) and running the appropriate decoding strategies (e.g., stochastic/top-k decoding).

### C.2  PREFIX INITIALIZATION

For the behavior LM to generate trajectories, an initial state must be provided so that the behavior LM can begin its generative process. Each trajectory represents a respective text sentence, and therefore every trajectory must be initialized with a different initial state. In light of this, we fed the behavior LM with sequences that were initialized with prefixes. Specifically, the first $p$ words of each text, $x_{1:p}$, were given as the initial state of the trajectory from which the behavior LM starts decoding (generative) process.

---

[3] If the intermediate actions $a_t$ and states $s_t$ are not considered but only the trajectory $\tau$, then we need to remove $\pi_\theta$'s influence on $\tau$ (e.g., $Q^{\pi_\theta}(s, a)$ and $a = \pi_\theta(s)$) because $\tau$ should be determined only by $\beta_{\bar{\theta}}$ in an off-policy setup.

### C.3 HYPERPARAMETERS

To fine-tune the target LM, we set the batch size, training epoch, learning rate $\alpha$, prefix length $p$, and total generation length $T$ to 256, 20, 5e-04, 2, and 15, respectively. Note that for computational efficiency, we randomly sampled around 50k samples from each dataset rather than using the full samples.

### C.4 PSEUDO ALGORITHM

---

**Algorithm 1** Off-policy policy gradient with reward dropout

---

1: Input: sequence data $x$, label data $y$, prefix length $p$, total length $T$, learning rate $\alpha$, dropout $\in$ {random, quantile}, dropout rate $\gamma \in [0, 1)$
2: Load pre-trained LLM as a behavior policy $\beta(\cdot)$ with parameter $\bar{\theta}$
3: Initialize target policy $\pi(\cdot)$ with parameter $\theta = \bar{\theta}$
4: Load pre-trained classifier as a reward model $R(\cdot)$ with parameter $\phi$ and fine-tune it on labels $y$
5: **for** epoch **do**
6:     $\tau \sim \beta_{\bar{\theta}}(\hat{x}_{p+1:T}|x_{1:p})$              ▷ generate trajectories from the prefix of $p$ length
7:     $\hat{r} = R_\phi(\tau)$                  ▷ calculate rewards of generated trajectories
8:     **if** dropout = random **then**
9:         $\hat{r}_{\text{dropout}} = \text{Random\_Dropout}(\hat{r}, \gamma)$   ▷ dropout $\gamma\%$ of rewards by random per batch
10:     **else if** dropout = quantile **then**
11:         $\hat{r}_{\text{dropout}} = \text{Quantile\_Dropout}(\hat{r}, \gamma)$     ▷ dropout bottom $\gamma\%$ of rewards per batch
12:     **else**
13:         $\hat{r}_{\text{dropout}} \leftarrow \hat{r}$                  ▷ no dropout
14:     **end if**
15:     $\nabla_\theta J_{\beta_{\bar{\theta}}}(\pi_\theta) = \mathbb{E}_{\tau \sim \beta_{\bar{\theta}}}\left[\nabla_\theta \pi_\theta(\tau) \times \hat{r}_{\text{dropout}}\right]$   ▷ calculate gradients of the target policy
16:     $\theta_{\text{new}} \leftarrow \theta + \alpha \nabla_\theta J_{\beta_{\bar{\theta}}}(\pi_\theta)$       ▷ update parameters of the target policy
17: **end for**
18: **return** optimal target policy parameters $\theta^*$

---

## D DATASET SUMMARY

| Dataset | | sentiment (Zhang et al., 2015) | politeness (Madaan et al., 2020) | toxicity (Dataset, 2017) | emotion (Li et al., 2017a) | topic (Zhang et al., 2015) |
|---|---|---|---|---|---|---|
| Data size (# of samples) | | 560,000 | 1,121,980 | 159,571 | 76,052 | 120,000 |
| # of labels (# of codes) | | 2 | 10 (2) | 2 | 7 (6) | 4 |
| Per label size | label 0 | 280,000 | 78,843 | 144,277 | 62,357 | 30,000 |
| | label 1 | 280,000 | 120,104 | 15,294 | 691 | 30,000 |
| | label 2 | | 123,942 | | 247 | 30,000 |
| | label 3 | | 95,333 | | 123 | 30,000 |
| | label 4 | | 83,860 | | 10,253 | |
| | label 5 | | 82,429 | | 877 | |
| | label 6 | | 88,274 | | 1,504 | |
| | label 7 | | 100,103 | | | |
| | label 8 | | 129,603 | | | |
| | label 9 | | 219,489 | | | |

Table 3: **(Descriptive Statistics)** The table shows the total number of samples, labels, and samples per label for each dataset. Note that the numbers in parentheses indicate the number of labels we used in training. For example, we dichotomized the 10 politeness labels by relabeling labels 0 to 8 as "non-polite", and we removed label 0 which denotes "no emotion" from the emotion dataset. Note that the gray cell highlights the severe imbalance in emotion labels.

# E    VISUALIZATION OF REWARD GROWTH

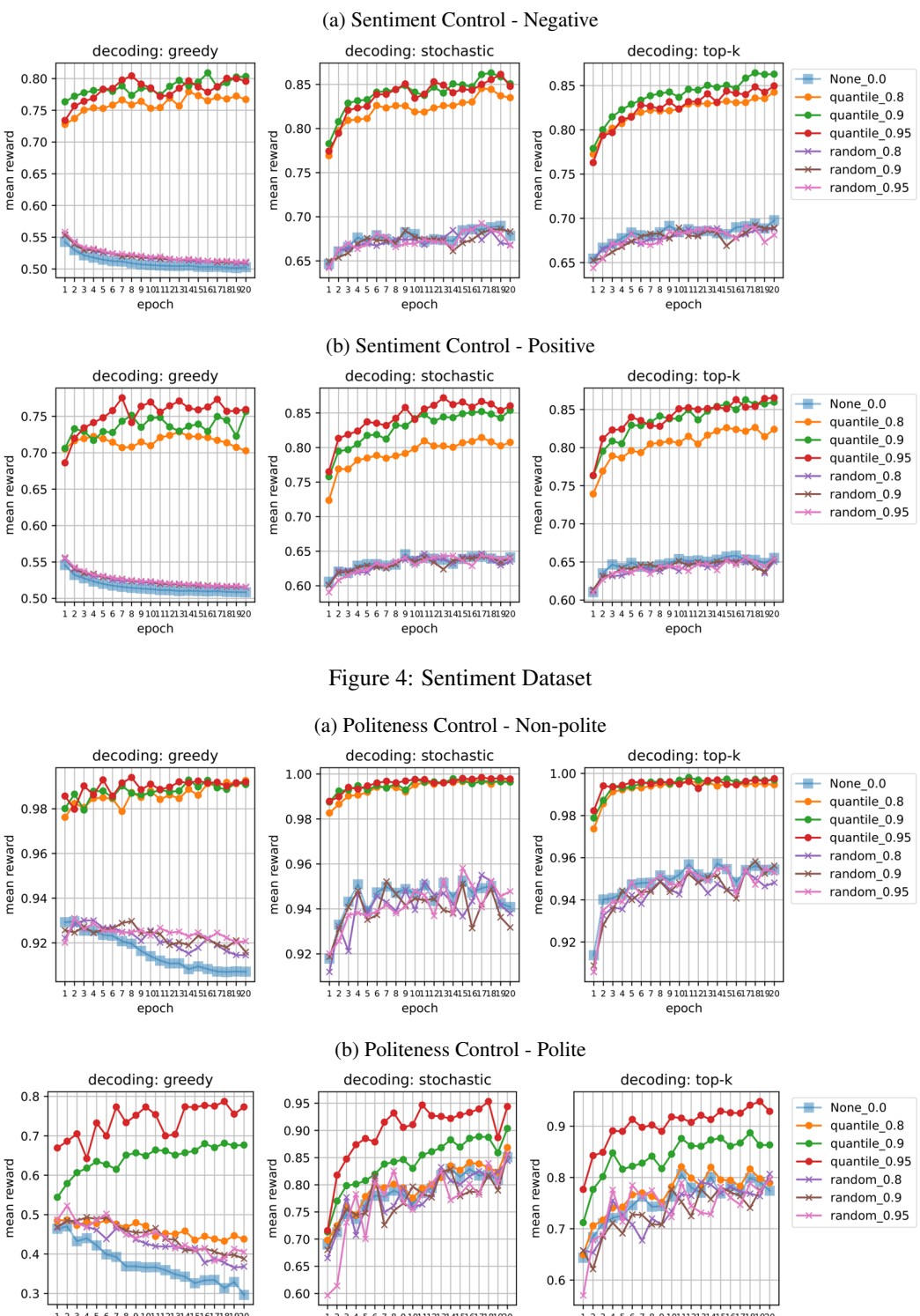

Figure 4: Sentiment Dataset

Figure 5: Politeness Dataset

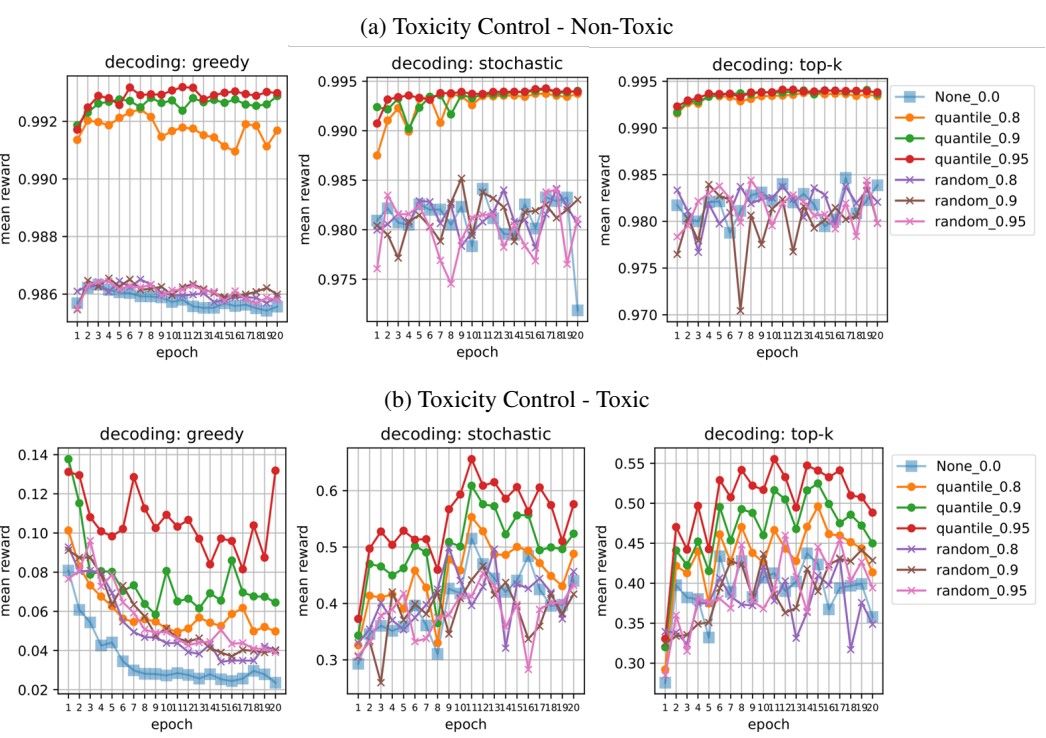

Figure 6: Toxicity Dataset

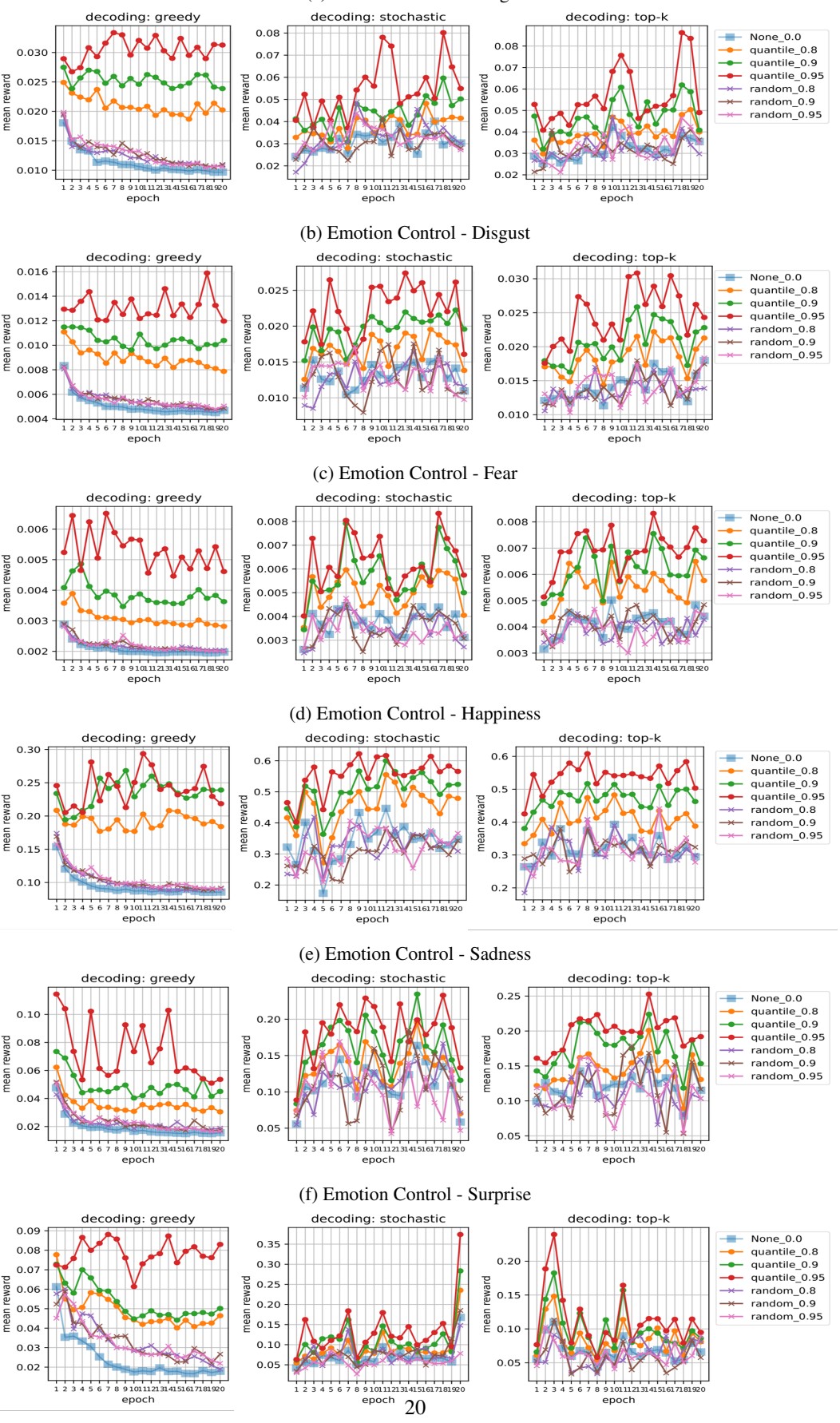

Figure 7: Emotion Dataset

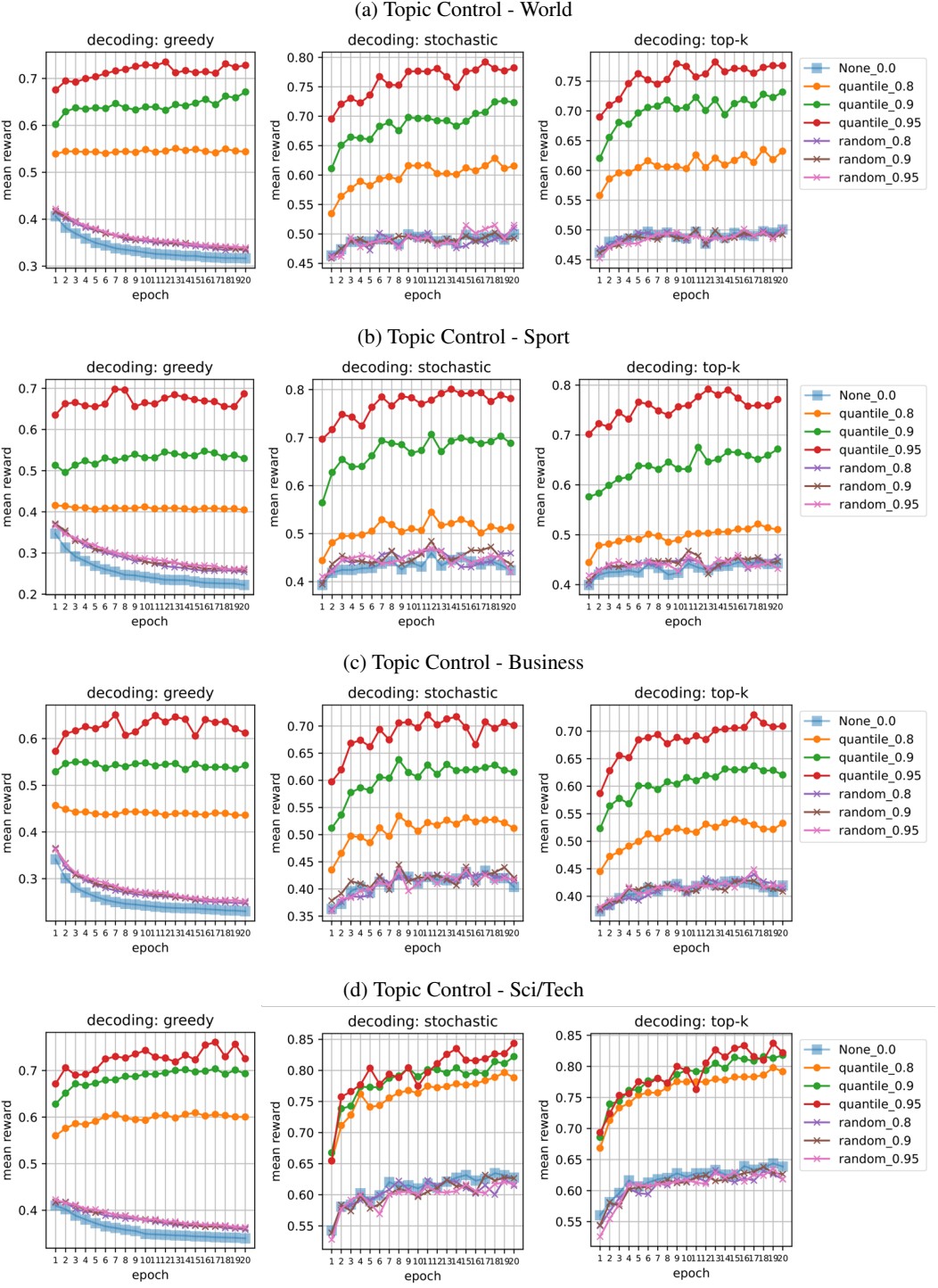

Figure 8: Topic Dataset

# F   GENERATED TEXT WITH INTENDED ATTRIBUTES

| Dataset | Control attribute | Generated text |
|---------|-------------------|----------------|
| *sentiment* | *negative* | The painting of the World, by Paul Thomas Woodford. |
| | | The chicken-crap, which is the worst thing I've ever seen. |
| | | The country's leaders have been accused of being using "toxic" |
| | *positive* | The painting is a beautiful, unique and unique collection of antique pieces from the British period. |
| | | The chicken is so delicious, it's a big one. |
| | | The country is so amazing, I'm going to do it!" |
| *politeness* | *non-polite* | I do not know that the same thing happened to Mr. |
| | *polite* | I do not know if you would like to see more of the new music. |
| *toxicity* | *non-toxic* | What is the most important thing that I've written? |
| | *toxic* | What the hell is wrong with that? |
| *emotion* | *anger* | When I hear that news, it's not like you're going to do this. |
| | *disgust* | When I hear that news, my wife's head was spinning. |
| | *fear* | When I hear that news, it's a shame. |
| | *happiness* | When I hear that news, it's a lot of fun. |
| | *sadness* | When I hear that news, it's a bit of an odd feeling. |
| | *surprise* | When I hear that news, it's a little bit strange. |
| *topic* | *world* | The issue focused on the fact that Iran is not a state of war, and it has been unable to defend its people. |
| | *sport* | The issue focused on the defense, which is a big part of what we have seen in recent years. |
| | *business* | The issue focused on the economy, but it also includes a number of other factors that have contributed to growth in GDP growth. |
| | *sci/tech* | The issue focused on the development of a new system for computing and networking is that it takes more than two seconds to develop. |

Table 4: **(Examples of Controlled Text)** The table shows examples of generated text that was controlled to have a specific attribute. During generation, the target LM was executed using top-k decoding ($k = 10$) and temperature sampling (temperature = 0.3). Note that underlined texts indicate the initialized prefixes. Some of the prefixes (e.g., "The painting", "The chicken", "The country", and "The issue focused on") are borrowed from existing literature (Dathathri et al., 2019), while the others are our own curation.

# G HUMAN EVALUATION

**Survey Design.** We planned a survey for human evaluation as shown in Figure 9. The survey was designed to include three types of questions. The first type includes questions that require participants to distinguish between real text and AI-generated text. This type was designed to measure how acceptable the generated texts are to humans. Similarly, the second type requires participants to distinguish if a text is real or generated. The only difference is that participants contrast real and generated texts (but which one is real is unknown to participants), then selects one that is likely to be written by a human. This type was designed to measure human-likeness of the generated text. Lastly, the third type requires participants to label appropriate control attributes (control codes) to the generated texts. This type was designed to test whether the improvement in control performance quantified in Table 1 and Appendix E meets human standards.

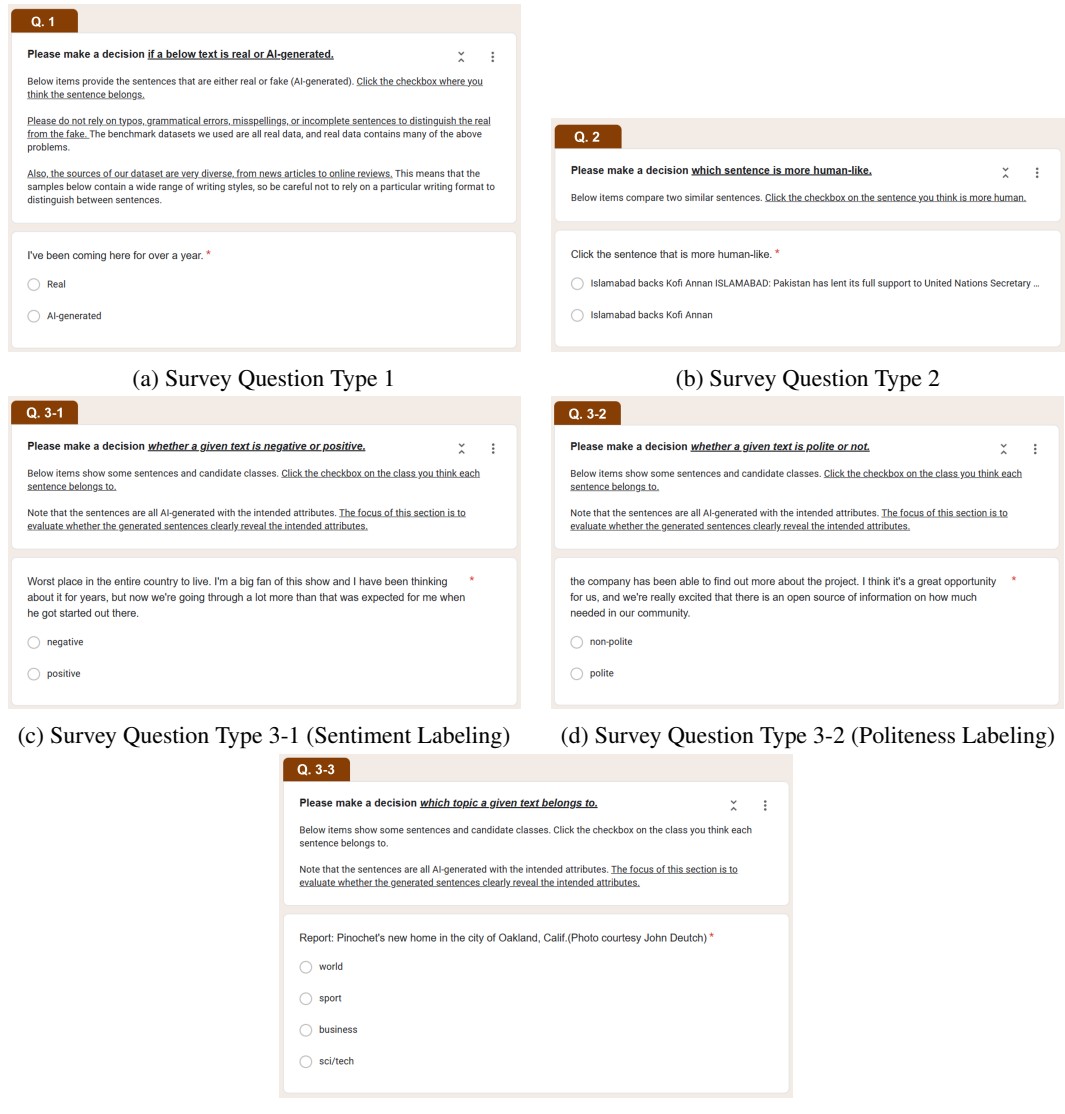

(a) Survey Question Type 1

(b) Survey Question Type 2

(c) Survey Question Type 3-1 (Sentiment Labeling)

(d) Survey Question Type 3-2 (Politeness Labeling)

(e) Survey Question Type 3-3 (Topic Labeling)

Figure 9: **(Survey Form.)** A total of 55 items were presented to participants where each item is categorized into one of three question types. As of the third type, we experimented only with *sentiment*, *politeness*, and *topic* datasets.

**Survey Analysis.** The percentage of correct answers by question type is summarized in the figure below. According to this figure, less than half of the respondents who answered Type 1 and 2 questions correctly. Considering that Type 1 and 2 questions ask participants if they can distinguish between real and generated texts, this result suggests that the controllable language model with reward dropout can generate reliable sentences. Meanwhile, more than half of the respondents replied with the correct answer on Type 3 question. Especially, over 70% of the respondents correctly labeled the sentiment and topic-control texts. This means that the performance improvements driven by reward dropouts reached human standards to some extent. Taking these all together, we can conclude that a target LM trained with SPG and reward dropout is able to generate reliable sentences while achieving a control performance in line with human standards. This implies that target LMs are unlikely to sacrifice likelihood objectives for reward improvements, i.e., it is likely to maximize the likelihood and reward objectives simultaneously.

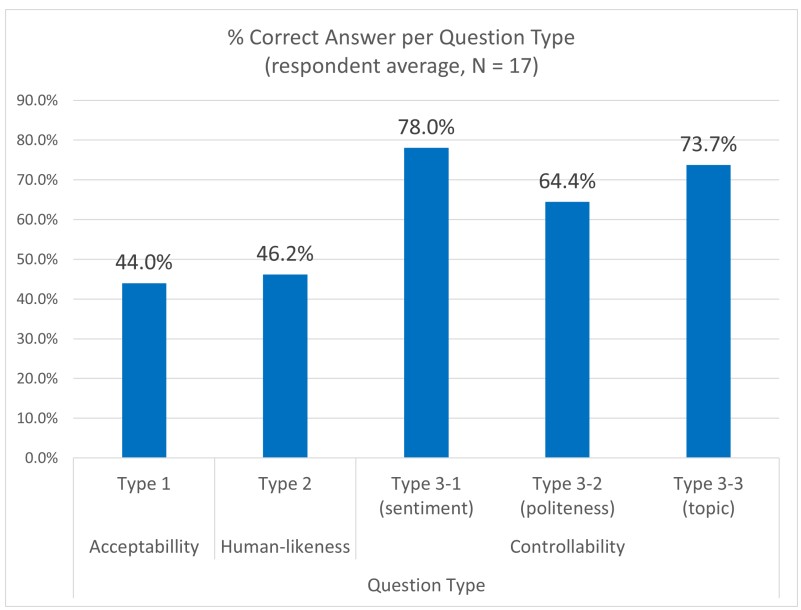

Figure 10: **(Survey Result.)** The survey was conducted with a total of 17 respondents. The respondent group was organized to include as diverse ethnicities (i.e., White, Hispanic, Mixed, East and Central Asian), genders (i.e., male and female), and ages (i.e., from 20 to 58) as possible.

# H    ADDITIONAL THEORETICAL ANALYSIS

In Pareto optimization, it is common that objectives are introduced as deterministic functions. However, in the context of RLM, a likelihood objective is represented by the behavior LM, and several pre-trained LMs can optionally be used; that is, different behavior policies (different likelihood objectives) can be considered. Accordingly, it would be useful to analyze how changes in behavior policy affect the optimal policy. To this end, we analyze two types of behavior policies here: high-informative behavior policy and ill-defined behavior policy, whose properties are described by Propositions H.1 and H.2, respectively:

**Proposition H.1** (Auxiliary Condition). *If* $\mathbb{E}_{\tau^*\sim\pi^*}[R(\tau)] = -\mathbb{E}_{\tau^*\sim\pi^*}[\ln\beta(\tau)]$ *holds, then* $\mathbb{E}_{\tau^*\sim\pi^*}[R(\tau)] = \mathcal{H}[\beta(\tau)]$ *holds proportionally.*

**Proposition H.2** (Violation Condition). *If* $\mathbb{E}_{\tau\sim\pi}[R(\tau)] \leq KL[\pi(\tau)||\beta(\tau)]$ *and* $\forall\tau\sim\pi,\ \beta(\tau)=0$ *hold, then the optimal policy* $\pi^*(\tau)$ *becomes a uniform policy.*

Proposition H.1 describes an auxiliary condition that leads to a further Pareto improvement: the higher the entropy of the behavior policy, the higher the maximal expected reward. Since entropy means information, it trivially states that *"a highly informative behavior policy increases the maximal level of expected reward."* That is, Proposition H.1 refers to a condition that improves the Pareto optimality itself, which is different from Theorem 4.3 that refers to a condition for an improved solution under the given Pareto optimality. The proof is provided in Appendix H.1.

On the other hand, Proposition H.2 implies that *"if the behavior policy is ill-defined, the optimal state of the target policy collapses into a completely random state."* When we say a behavior policy is ill-defined, it means that the optimal policy does not exist with that behavior policy. In other words, if a behavior policy violates Eq (6) under a certain condition, that condition is a violation condition and yields an ill-defined behavior policy. In Appendix H.2, we prove that the violation condition is given by $\forall\tau\sim\pi,\ \beta(\tau)=0$, in which case the optimal policy will be a uniform policy.

## H.1    PROOF OF PROPOSITION H.1

**Proposition I.1**    *If* $\mathbb{E}_{\tau^*\sim\pi^*}[R(\tau)] = -\mathbb{E}_{\tau^*\sim\pi^*}[\ln\beta(\tau)]$ *holds, then* $\mathbb{E}_{\tau^*\sim\pi^*}[R(\tau)] = \mathcal{H}[\beta(\tau)]$ *holds proportionally.*

*Proof.* Suppose the Pareto optimality is given by $\mathbb{E}_{\tau^*\sim\pi^*}[R(\tau)] = -\mathbb{E}_{\tau^*\sim\pi^*}[\ln\beta(\tau)]$. At the optimal points $\tau^*\sim\pi^*$, trivially, $\pi^*(\tau) = \beta(\tau)e^{R(\tau)}$ holds and $R(\tau)$ has maximal values for all $\tau$. Let $k$ be the maximal value of $R(\tau)$. Then, we can replace $\pi^*(\tau)$ by $\beta(\tau)e^k$ and arrange the right-hand side of Eq (11) by

$$-\mathop{\mathbb{E}}_{\tau^*\sim\pi^*}[\ln\beta(\tau)] = -\mathop{\mathbb{E}}_{\tau\sim\beta}[e^k\ln\beta(\tau)] = \alpha\mathcal{H}[\beta(\tau)] \propto \mathcal{H}[\beta(\tau)],$$

where $\alpha = e^k$ is a proportionality constant (e.g., if $k=1$ then $e^k\approx 2.718$.) and can be ignored. As a result, $\mathbb{E}_{\tau^*\sim\pi^*}[R(\tau)] = \mathcal{H}[\beta(\tau)]$ holds proportionally. $\square$

## H.2    PROOF OF PROPOSITION H.2

**Proposition I.2**    *If* $\mathbb{E}_{\tau\sim\pi}[R(\tau)] \leq KL[\pi(\tau)||\beta(\tau)]$ *and* $\forall\tau\sim\pi,\ \beta(\tau)=0$ *hold, then the optimal policy* $\pi^*(\tau)$ *becomes a uniform policy.*

*Proof.* Let a target policy have a positive range $\pi(\tau)\in(0,1]$, or equivalently, $0<\pi(\tau)\leq 1$. This condition is stricter but reasonable since we are only dealing with sampled (feasible) trajectories $\tau\sim\pi$, and the sampled trajectories represent non-zero probabilities. Next, let us consider the optimal state $\pi(\tau) = \beta(\tau)e^{R(\tau)}$. Considering the optimal state is required not only because it is a necessary condition for the reward upper bound $\mathbb{E}_{\tau\sim\pi}[R(\tau)] \leq KL[\pi(\tau)||\beta(\tau)]$, but also because it states the existence of an optimal solution.

Considering $\pi(\tau)\in(0,1]$ and $\pi(\tau) = \beta(\tau)e^{R(\tau)}$ together, the domain of target policy $\tau\in\mathcal{T}_\pi$ is defined by which $\beta(\tau)e^{R(\tau)}\in(0,1]$ is satisfied. This implies that the behavior policy $\beta(\tau)$ and the reward objective $R(\tau)$ are well-defined only in that domain, having a support set given by

$\{\exists \tau \in \mathcal{T}_\pi \mid 0 < \beta(\tau)e^{R(\tau)} \leq 1\}$. Conversely, out of that domain, e.g., $\beta(\tau)e^{R(\tau)} = 0$,[4] a support set is given by $\{\forall \tau \in \mathcal{T}_\pi \mid \beta(\tau)e^{R(\tau)} = 0\}$, implying that either $\beta(\tau)$ or $R(\tau)$ is ill-defined. Since $e^{R(\tau)}$ is positive for all $R(\tau) \in [0,1]$, trivially $\beta(\tau)e^{R(\tau)} = 0$ holds if and only if $\beta(\tau)$ is zero for all $\tau$, i.e., $\forall \tau, \; \beta(\tau) = 0$.

Now, suppose $\mathbb{E}_{\tau \sim \pi}[R(\tau)] \leq \mathrm{KL}[\pi(\tau)||\beta(\tau)]$ and $\forall \tau \sim \pi, \; \beta(\tau) = 0$ holds. Then, the reward changes to the entropic reward, and the reward upper bound disappears as $-\ln \beta(\tau) \to +\infty$ at $\beta(\tau) = 0$.

$$\underbrace{\mathbb{E}_{\tau \sim \pi}[R(\tau)] \leq - \mathbb{E}_{\tau \sim \pi}[\ln \beta(\tau)] - \mathcal{H}[\pi]}_{= \text{ Reward Upper Bound}} \iff \underbrace{\mathbb{E}_{\tau \sim \pi}[R(\tau)] + \mathcal{H}[\pi]}_{= \text{ Entropic Reward}} \leq \underbrace{- \mathbb{E}_{\tau \sim \pi}[\ln \beta(\tau)]}_{= +\infty \;\; \text{s.t.} \;\; \beta(\tau) = 0}$$

As a result, the bounded reward maximization turns into an unbounded entropic reward maximization, and thus the optimal policy becomes a uniform policy as it has the highest entropy.  $\square$

### H.3  BEHAVIOR POLICY IN 10-TURN POSITIONING GAME

In the 10-turn position game, we implemented the behavior policy (a policy of the behavior agent) based on the truncated normal distribution according to Burkardt (2014). Note that a support set of the normal distribution is defined on a real-value domain, but the action space (and thus position space) must be an integer domain. Accordingly, we integerized it by rounding up if the sampled action is greater than the average and rounding down if it is less. For example, assume $\mu$ and $\sigma$ of the behavior policy are 3 and 0.5, respectively, and suppose one sampled action by the behavior policy is 3.71. In this case, we round up 3.71 by 4, and thus the next visiting state is set to 4. Similarly, if a sampled action is 3.12, then the next visiting state is set to 3.

### H.4  EMPIRICAL VALIDATION OF PROPOSITIONS

This section provides an empirical analysis that demonstrates all theoretical results presented in the previous section. To this end, we devised a simulation experiment called a 10-turn positioning game. The goal of this experiment is to confirm the theoretical results and analyze them in the RLM context.

**10-turn Positioning Game.**  Figure 11a describes the 10-turn positioning game. In this game, an agent changes its position over 10 turns. Each turn is indexed by $t = 1, ..., 10$. Two agents participate in this game: the behavior agent and the target agent. Each agent selects one of 10 actions $a_t \in [1, 2, ..., 10]$ in each $t$-th turn and moves to a corresponding position $i \in [1, 2, ..., 10]$. The history of an agent's position is a trajectory $\tau$. The behavior agent changes its positions based on a normal distribution policy, $a_t \sim \mathcal{N}(\mu, \sigma^2)$ for $t = 1, ..., 10$ and collects rewards if the occupied position is rewarded.[5] The target agent observes a trajectory of the behavior agent and learns from it in an off-policy manner. For simplicity, we set a reward distribution to have an exponential shape only at $i \in [6, 10]$. Note that agent's state $s_t$ is defined as a vector, which describes a cumulative visit frequency to each position, $\forall t, \; s_t = [s_1, \cdots, s_i, \cdots, s_{10}]$, and $\forall i, \; s_i \in \mathbb{Z}_0^{10}$; here, $s_i \in \mathbb{Z}_0^{10}$ indicates that the value of $s_i$ is defined by integer numbers between 0 and 10. The state vector is initialized to a random position at the first turn.

**Simulation Results & Interpretations.**  Figure 11b shows the simulation results. A total of 5000 trajectories were simulated ($N = 5000$) where each trajectory has a length of 10 ($L = 10$) and reward values were normalized by the softmax function. The rewards were distributed to each position in ascending order from smallest to largest. That is, the largest reward is 1.0 at position 10, implying the expected reward of the target agent, i.e., $\mathbb{E}_{\tau \sim \pi}[R(\tau)] = \frac{1}{N \times L} \sum_{n=1}^{N} \sum_{t=1}^{L} R(s_t^n, a_t^n) \triangleq \tilde{r}$, can be up to 1.0 at maximum.

The first and second columns in Figure 11b demonstrate Propositions H.1 and H.2, respectively. The first column shows that the more uniform the behavior agent's policy, the higher the target agent's expected reward (Cases 0, 1 and 2). This result is consistent with Proposition H.1. On the

---

[4]We do not consider $\beta(\tau)e^{R(\tau)} > 1$ for the out-of-domain because there are infinitely many combinations of $\beta(\tau) \in [0,1]$ and $R(\tau) \in [0,1]$ that satisfy $\beta(\tau)e^{R(\tau)} > 1$.

[5]Refer to Appendix H.3 for the details.

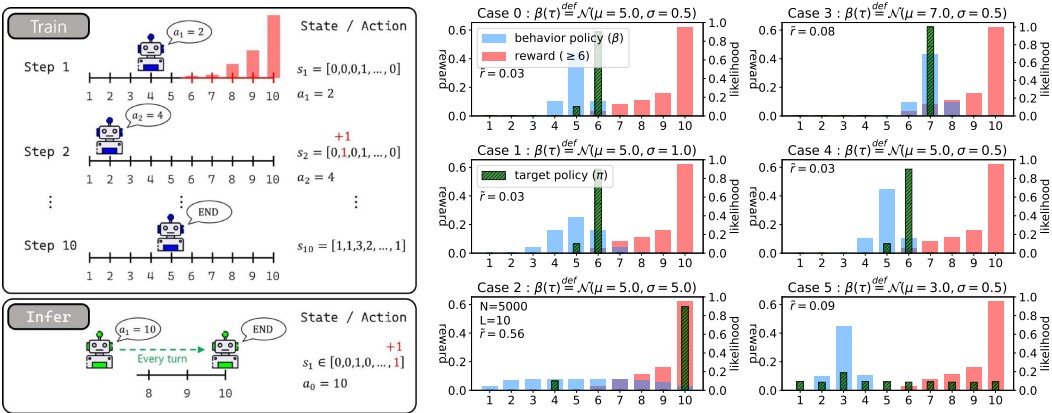

(a) **Concept Illustration:** The horizontal bar shows available positions and actions together. Red positions indicate a reward zone. Blue/green agents refer to behavior/target agents, respectively. Note that 10 turns (steps) make up a single trajectory.

(b) **Results:** The blue and green bars represent the behavior/target agents' visit frequency to corresponding positions. The red bar indicates a reward distribution defined over $i \in [6, 10]$. $\tilde{r}$ is the expected reward achieved by a target agent. (1) In cases 0-4, $\pi$ converges to only a few actions that maximize reward. (2) In case 5, $\pi(\tau)$ collapses into a uniform policy because $\beta(\tau)$ is ill-defined.

Figure 11: (**10-turn Positioning Game.**) Concepts and Results

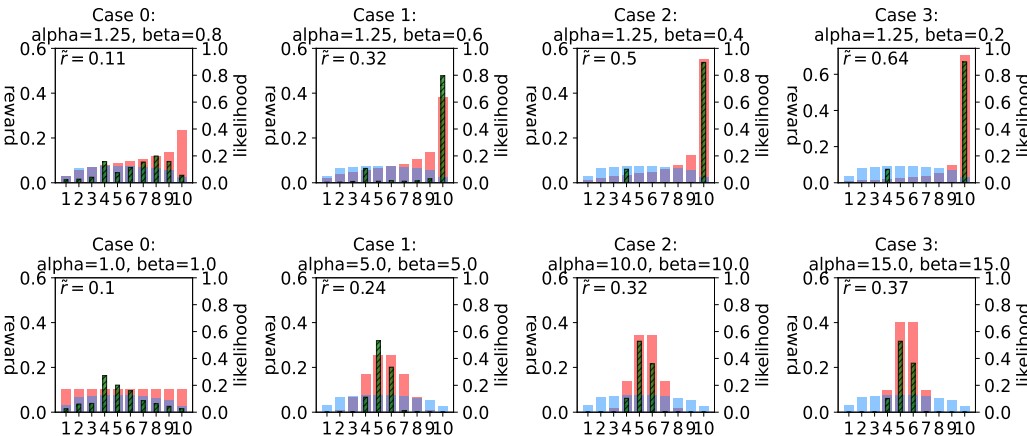

Figure 12: (**Reward Manipulation & Pareto Improvement.**) The behavior and target agents' policies are represented by the blue and green bars, respectively. The behavior policy was set to a normal distribution whose parameters are $\mu = 5.0$ and $\sigma = 5.0$, and each manipulated reward distribution was defined over full positions $i \in [1, 10]$ and set to a beta distribution for different shape parameters alpha and beta. Other conditions were set the same as before (e.g., $N = 5000$, $L = 10$).

other hand, the second column shows that the farther the behavior agent's policy is defined from the reward distribution, the less reward the target agent collects (Cases 3 and 4). Furthermore, if the behavior agent's policy is ill-defined (i.e., if the behavior agent cannot enter the reward zone, or the behavior agent samples no actions between 6 and 10.), then the target agent receives no rewards from the behavior agent. As a result, the target agent's policy converges to a uniform policy, maximizing entropic rewards (Case 5). This result is consistent with Proposition H.2.

We can also interpret Figure 11b from the RLM perspective. The first column highlights that target LMs will be better controlled if behavior LMs can cover as large a token space (a dictionary) as possible. This is because the large token space is more likely to create opportunities for exploring higher-reward sentences. Therefore, it is recommended to use large language models (LLMs) when building RLMs. The second column emphasizes the importance of preparing the training dataset

correctly. For example, let us say we need to control the target LM so that sentences are generated with negative sentiment. If the behavior LM is pre-trained on a dataset consisting only of positive sentences (i.e., the behavior policy is ill-defined), then the behavior LM cannot provide negative candidate sentences. Consequently, the target LM cannot experience any rewards and therefore never be controlled.

## H.5   Non-normal Behavior Policy

In this subsection, we provide additional results when a behavior policy is non-normal.

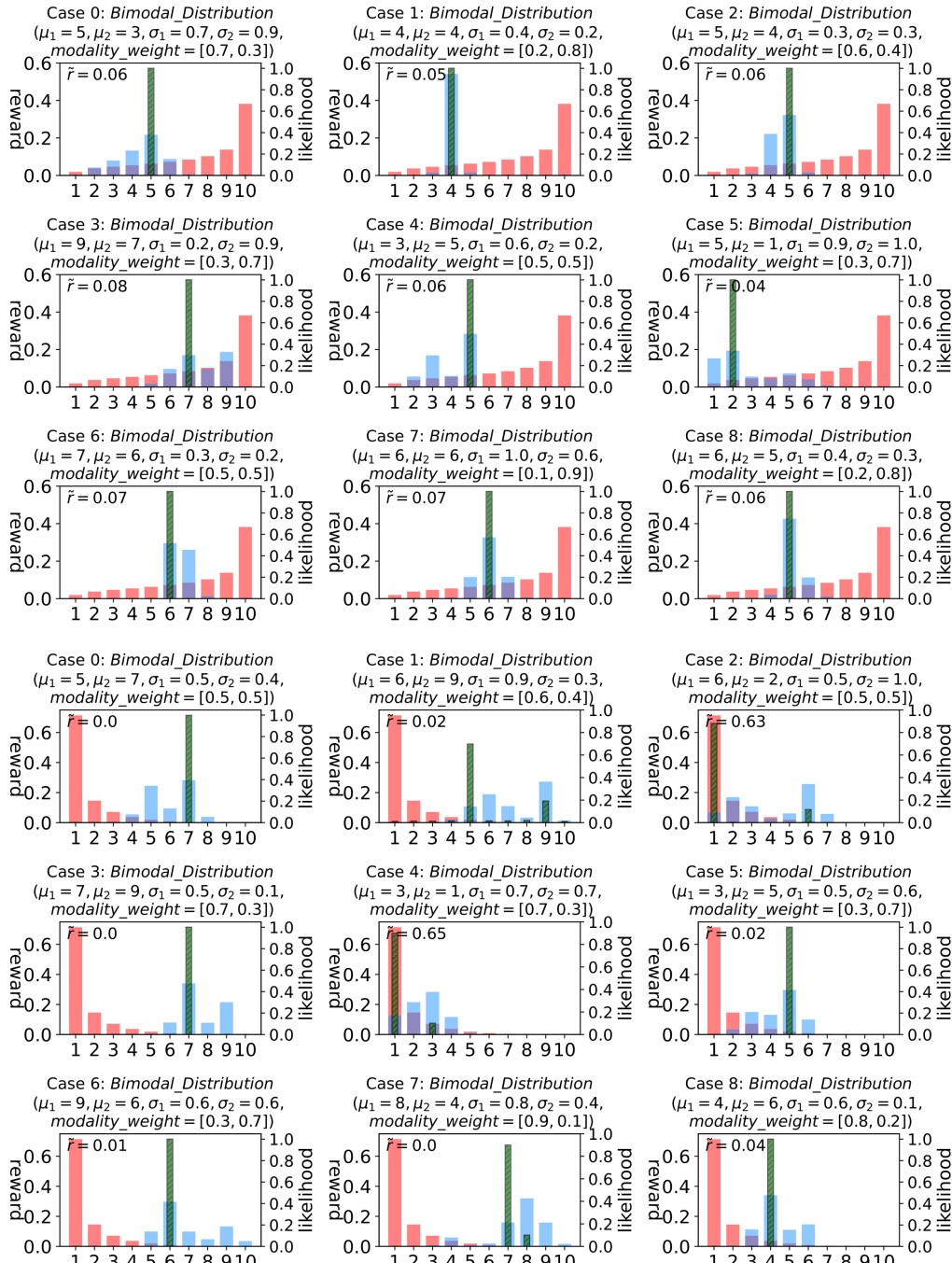

Figure 13: Additional results with non-normal behavioral policies and different reward distributions.

