# OpenReview forum: "A bi-objective perspective on controllable language models: reward dropout improves off-policy control performance"
_ICLR.cc/2024/Conference — Submitted to ICLR 2024_

### Official Review · Reviewer_dnJ6 · 2023-10-31

**Soundness:** 2 fair
**Presentation:** 3 good
**Contribution:** 4 excellent
**Rating:** 8
**Confidence:** 3

**Summary:**

This work studies a trade-off between reward and likelihood, which is an important but unexplored problem in the pertaining or finetuning of LMs. And the authors proposed a simple solution to this problem, dubbed Reward Dropout.

**Strengths:**

The problem studied in this work is interesting and important, and the authors provide a thorough theoretical analysis from the perspective of Pareto optimization/bi-objective optimization.

**Weaknesses:**

This work only focuses on a single "balanced" Pareto solution to the proposed bi-objective optimization, which weakens the motivation for using bi-objective formulation. According to the experimental results such as Fig. 3, if you only consider the reward metric and would like to relatively neglect the likelihood objective, why not consider the trade-off problem from the perspective of constrained optimization with the likelihood objective as the constraint?

One possible solution is that the authors can provide evidence reflecting some other trade-off solutions on the Pareto front are also important (e.g., plotting an approximate Pareto front and showing different behaviors of Pareto solutions).

Moreover, the experiment only uses a relatively small language model (i.e., GPT-2). LLMs can weaken the influence of the reward-likelihood trade-off due to their larger model capacity.

**Questions:**

Some comments:

1. for eq. (5), if the behavior policy has already maximized the reward objective, will the bi-objective optimization reduce to a single objective optimization?

2. for fig. 2, please provide additional results under a non-normal distribution behavior policy.

3. What is the core idea of reward dropout? I think the core idea is to relax the distribution of rewards in order to achieve an easier Pareto improvement. From this perspective, I wonder why the final performance is improved by the quantile dropout that sharpens the distribution.

---post-rebuttal comment---

After reading the authors' responses and their revised version, I decided to raise my score.

---

> ### Author Response · Authors · 2023-11-20
>
> ### Weaknesses
> > 1. *“This work only focuses on a single "balanced" Pareto solution to the proposed bi-objective optimization, which weakens the motivation for using bi-objective formulation. According to the experimental results such as Fig. 3, if you only consider the reward metric and would like to relatively neglect the likelihood objective, why not consider the trade-off problem from the perspective of constrained optimization with the likelihood objective as the constraint? One possible solution is that the authors can provide evidence reflecting some other trade-off solutions on the Pareto front are also important (e.g., plotting an approximate Pareto front and showing different behaviors of Pareto solutions).”*
>
> ---
>
> - Dear Reviewer #3, thank you for your thoughtful feedback. Personally, this comment really struck a chord with us - your question is the one that has the most implications for our research, which explores the "perspective" of training RLMs, as opposed to the majority of recent work that focuses on developing model, designing neural network structure, or implementing an algorithm. We are truly touched by your feedback.
> - As this was a very important question with implications for our study, we wrote an answer for this as *Meta Response 3 (MR3) to be shared with all reviewers and ACs.* Please refer to our response in *MR3*. We have also included additional experiments related to your question in Appendix I. Please see the revised submission.
>     - In the additional experiment, you can see that if the RLM is trained without *the transfer learning*, then the reward and likelihood rise together (i.e., maximizing both objectives simultaneously, which implies a bi-objective problem) until the reward touches RUBO, and then as training continues, the reward objective is sacrificed to maximize the likelihood objective (This is the theoretical Issue mentioned in MR3).
>     - The trend of the reward and likelihood objective increasing together until the reward touches the RUBO shows that RLM training is a bi-objective problem, not a constrained single-objective optimization problem.
>
> - Once again, thank you so much for your thoughtful feedback.
>
> ---
>
> > 2. *“Moreover, the experiment only uses a relatively small language model (i.e., GPT-2). LLMs can weaken the influence of the reward-likelihood trade-off due to their larger model capacity.”*
>
> ---
>
> - Dear Reviewer #3, thank you for your thoughtful feedback. If you are asking *"Does using LLMs with greater capacity mitigate the reward-likelihood trade-off?"*, we would say *"yes"*.
>
> - In fact, we have already discussed this in Fig 1(b) and Section 5. Looking at the first column (left-most column) of Fig 1(b), we can see that the target policy $\pi$ (green bars) is skewed to the right, where the higher reward exists from Case 0 to Case 2.
>
> - Here, the green bar represents the position to which $\pi$ converges, and it implies the reward upper bound (RUBO) by which $\pi$ can maximize the reward subject to the behavior policy $\beta$ (blue bars) given for each case.
>
> - To summarize, Fig 1(b) shows that the wider the coverage of $\beta$, i.e., the larger the capacity of the LLM, the higher the RUBO, and thus the position of the green bar shifts to the right. In other words, a larger capacity of LLMs mitigates the reward-likelihood trade-off and increases controllability.
>
> - Meanwhile, thanks to your feedback, we realized that it would be very instructive to experiment with Reward Dropout for different LLMs and compare their effects directly. Therefore, we have conducted additional experiments. Please refer to Appendix J. This experiment compared reward dropout effects across GPT2 (117M), OPT (350M), XGLM (564M), and GPT2-Large (774M). It was your thoughtful feedback that inspired us to conduct this experiment. We would like to express our sincere gratitude.

---

> ### Author Response · Authors · 2023-11-20
>
> ### Questions
> > 1. *“for eq. (5), if the behavior policy has already maximized the reward objective, will the bi-objective optimization reduce to a single objective optimization?”*
>
> ---
>
> - Dear Reviewer #3, thank you for your thoughtful feedback. our answer to your question is *"yes"*. It is correct that the bi-objective problem reduces to single-objective optimization. We think it can be easily explained with a 10-turn positioning game.
> - Let us think of Fig 1 or Fig 2. If the behavior policy $\beta$ is already maximizing $R(\tau)$, then the distribution of $\beta(\tau)$ (the blue bars) will be skewed towards the highest rewarding position. This is because if the behavior trajectories $\tau \sim \beta$ are the optimal trajectories that maximize $R(\tau)$, there are no positions from which $\tau$ is sampled except for the highest reward positions.
> - This means that the target policy $\pi$ will be trained with only trajectories of the highest reward position, where reward maximization is already guaranteed by $\beta(\tau)$. Accordingly, all that $\pi$ needs to optimize is the likelihood objective, i.e., $\pi$ only needs to converge to the behavior policy $\beta$.
>
> ---
>
> > 2. *“for fig. 2, please provide additional results under a non-normal distribution behavior policy.”*
> - Dear Reviewer #3, thank you for your thoughtful feedback. We sincerely appreciate your request for additional experiments on the main content of our study. Due to a lack of space in the main text, the requested experiment has been included in Appendix C.2. Please refer to it.
> - In addition to the results presented in Appendix C.2, we conducted more experiments with different behavior policies and reward distributions. However the message from the 10-turn positioning game is consistent: *“To maximize rewards, the coverage of a behavior policy should be broad enough to explore high rewarded areas, i.e., the use of LLMs is highly encouraged.”* Hence, we have included only a few of the experiments in Appendix C.2 as examples.

---

> ### Author Response · Authors · 2023-11-20
>
> ### Questions (cont.)
> > 3. *“What is the core idea of reward dropout? I think the core idea is to relax the distribution of rewards in order to achieve an easier Pareto improvement. From this perspective, I wonder why the final performance is improved by the quantile dropout that sharpens the distribution.”*
>
> - Dear Reviewer #3, thank you for your thoughtful feedback. You are correct that the core idea of Reward Dropout is to relax the reward distribution in order to achieve Pareto improvement. To be more precise, Reward Dropout is a technique to *"manipulate the reward distribution to force the update of $\pi$ toward a direction where Pareto improvement is guaranteed"*.
> - Here we need to decide *"which way to manipulate"* the reward. You may be wondering at this point why it has to be a "quantile" dropout. To explain this, we first need to understand Theorems 4.1, 4.2, and 4.3, and how they lead to the Reward Dropout:
>     - Based on Theorems 4.1 and 4.2, we proved that Eq (4) contains an inherent tradeoff between the likelihood objective $\beta(\tau)$ and the reward objective $R(\tau)$. Specifically, the RUBO in Theorem 4.1 and the log-negative relation in Theorem 4.2 clearly demonstrate that there exists a trade-off between $\beta(\tau)$ and $R(\tau)$ in Eq (4). Therefore, we can assure that the problem of minimizing Eq (4) is equivalent to a Pareto optimization problem that optimizes two conflicting objectives.
>         - $\text{Eq (4)} \qquad \text{KL}\left[ \pi(\tau) || \beta(\tau) e^{R(\tau)} \right] = \sum_{\tau} \pi(\tau) \ln\frac{\pi(\tau)}{\beta(\tau)e^{R(\tau)}}$
>         - $\text{(Theorem 4.1)} \qquad \text{KL}\left[ \pi(\tau) || \beta(\tau) e^{R(\tau)} \right] \Longrightarrow \mathbb{E}\_{\tau \sim \pi} \left[ R(\tau) \right] \leq \text{KL} \left[ \pi(\tau) || \beta(\tau) \right] = \text{RUBO}$
>         - $\text{(Theorem 4.2)} \qquad \mathbb{E}\_{\tau \sim \pi} \left[ R(\tau) \right] \leq \text{KL} \left[ \pi(\tau) || \beta(\tau) \right] \ \overset{\pi \rightarrow \pi^{\*}}{\Longrightarrow} \ \mathbb{E}\_{\tau^{\*} \sim \pi^{\*}} [ R(\tau) ] = - \mathbb{E}\_{\tau^{\*} \sim \pi^{\*}} [ \ln \beta(\tau)] \ \Longleftrightarrow \ R(\tau)=-\ln\beta(\tau) , \ \ \forall{\tau^{\*}} \sim \pi^{\*}$
>         - In Meta Response 2 (MR2), we have explained more about how RUBO implies a trade-off. Please refer to MR2.
>
>     - **Both Eq (6) and Theorem 4.2 describe the optimal state of Eq (4)**, which implies a Pareto optimal solution $\tau^{\*} \sim \pi^{\*}$. This means that **Eq (6) and the result of Theorem 4.2 is just describing the same phenomenon,** i.e., the Pareto optimal state, **from the different perspective**, **one focusing on bi-objectiveness** (Eq (6)) **and the other on the trade-off** **between the two objectives** (Theorem 4.2).
>         - $\text{Eq (6)} \qquad \pi^{\*}(\tau)=\frac{\beta(\tau)e^{R(\tau)}}{e^{1-\lambda}} = \beta(\tau)e^{R(\tau)} \quad \left( \because \lambda = 1 \right)$
>         - As for why it should be $\lambda=1$, please refer to our response to Weakness 5 from reviewer #2.

---

> ### Author Response · Authors · 2023-11-20
>
> ### Question (cont.)
> > 3. *“What is the core idea of reward dropout? I think the core idea is to relax the distribution of rewards in order to achieve an easier Pareto improvement. From this perspective, I wonder why the final performance is improved by the quantile dropout that sharpens the distribution.”*
>
>   - **In a Pareto optimization problem,** it is obvious that **any policy that does not achieve Pareto optimality**, i.e., $\pi(\tau)\neq \beta(\tau)e^{R(\tau)}$, **always has room for Pareto improvement**. And we know that Pareto optimality presented in Theorem 4.2 is derived from RUBO presented in Theorem 4.1.
>     - Here, through *reductio ad absurdum* (i.e., through the proof by contradiction), **Theorem 4.3 shows that** the contraposition of Theorem 4.1, i.e., $\mathbb{E}\_{\tau \sim \pi}[R(\tau)] > \text{KL}[\pi(\tau)||\beta(\tau)]$, leads to$\pi(\tau)\neq \beta(\tau)e^{R(\tau)}$, or **the negation of Theorem 4.1 leads to the negation of Eq (6), and Pareto optimality does not hold.** In other words, **Theorem 4.3 provides the condition for the Pareto improvement.**
>         - $\text{(Theorem 4.3)} \qquad \mathbb{E}\_{\tau \sim \pi}[R(\tau)] > \text{KL}[\pi(\tau)||\beta(\tau)]  \ \overset{\text{Eq (14)}}{\Longrightarrow} \ \mathbb{E}\_{\tau \sim \pi}[R(\tau)] + \mathbb{E}\_{\tau \sim \pi}[\ln \beta(\tau)] > 0 \ \Longrightarrow \ \pi(\tau)\neq \beta(\tau)e^{R(\tau)}$
>
>     - Therefore, **by forcing the Pareto improvement condition given by Theorem 4.3 to hold**, i.e.,  $\mathbb{E}\_{\tau \sim \pi}[R(\tau)] > \text{KL}[\pi(\tau)||\beta(\tau)] \ \Longrightarrow \ \mathbb{E}\_{\tau \sim \pi}[R(\tau)] + \mathbb{E}\_{\tau \sim \pi}[\ln \beta(\tau)] > 0$, **we can ensure that the target policy $\pi$ or the target LM $\pi\_{\theta}$ is updated only in the direction of Pareto improvement** that increases both the likelihood and reward objectives without unnecessary exploration, and as a result reach the Pareto optimal solution faster.
>     - The message behind Theorem 4.3 is simple: *“the Pareto improvement condition is satisfied when $\mathbb{E}\_{\tau \sim \pi}[R(\tau)] + \mathbb{E}\_{\tau \sim \pi}[\ln \beta(\tau)] >0$, so manipulate either or both $R(\cdot)$ and $\beta(\cdot)$ to hold it.”* In other words, **having Eq (7) hold will always update the parameter $\theta$ of the target policy (target LM) $\pi\_{\theta}$ in the direction of Pareto Improvement.**
>         - $\text{Eq (7)} \qquad \mathbb{E}\_{\tau \sim \pi}[R(\tau)] + \mathbb{E}\_{\tau \sim \pi}[\ln \beta(\tau)] > 0$
> ---
>   - Given that the behavior policy is either a pre-defined distribution, i.e., $\beta(\cdot)$, or a pre-trained LM, i.e., $\beta_{\bar{\theta}}(\cdot)$, whose parameter is fixed as $\bar{\theta}$, **it is only $R(\cdot)$ that can be manipulated. Accordingly, we only need to focus on $R(\tau)$ to hold Eq (7).**
>     - Specifically, we need to **manipulate $R(\tau)$ such that only a few high rewards are considered** because $\mathbb{E}\_{\tau \sim \pi} \left[ R(\tau) \right]$ refers to the average reward $\tilde{r}$ and the average is sensitive to bias caused by outliers. Extremely saying, **a single high reward is more effective at satisfying the Pareto improvement condition, i.e., $\mathbb{E}\_{\tau \sim \pi}[R(\tau)] + \mathbb{E}\_{\tau \sim \pi}[\ln \beta(\tau)] > 0$, than many average rewards.**
>         - In fact, we have already written about this in the Reward Manipulation paragraph in Section 5. Please refer to the orange text in section 5 of the revised submission.
>
>     - This is where we can explain why quantile dropout is effective. **Quantile dropout is literally a way to discard all samples with rewards below a predefined quantile and consider only a few samples with higher rewards**. In other words, **it's a technique that reinforces the bias toward high-reward samples by focusing on highly rewarding outliers to achieve Pareto Improvement.** This is why Quantile Dropout cannot help but drive Pareto Improvement.
> ---
> - Please refer to Meta Response 2 (MR2) and Appendix B for more details related to Theorems 4.1, 4.2, and 4.3.

---

> ### Author Response · Authors · 2023-11-22
> **Thank you for raising your score.**
>
> Thank you so much, you helped us clarify some things and improve our contribution.
>
> We are very grateful to you for asking questions that are particularly relevant to the essence of our research, e.g., *"Why not consider constrained optimization?"*, *"What is the core idea of reward dropout"*, as well as pointing out trivial but easily missed experiments, e.g., *"LLMs can weaken the influence of the reward-likelihood trade-off due to their larger model capacity"*, which contributed greatly to our research.
>
> Your dedicated feedback and scores have proven to us that our research is valuable. Thank you from the bottom of our hearts. If you have any additional comments, please let us know.

---

### Official Review · Reviewer_EE6f · 2023-11-01

**Soundness:** 3 good
**Presentation:** 1 poor
**Contribution:** 2 fair
**Rating:** 5
**Confidence:** 2

**Summary:**

The paper provides some theoretical aspects of the RLMs by casting it as an offline RL problem. With the offline RL objective, the paper provides some properties of the optimal policies or pareto optimal policies. Based on these observations, the paper provides the reward dropout method to improve policies and tested on several benchmarks.

**Strengths:**

The reward dropout methods seem simple and widely applicable. The experiment results indicate the method is actually improving performance in practice. However, I do not conduct research on LLM at all so I am unable to judge the significance of the results from the experiment.

**Weaknesses:**

The technical section of the paper is poorly written and there are many questionable claims. For example,

1. The CML and RLM are still different problems. The paper should not claim that they analysis CLM by RLM because they are "intrinsically analogous".

2. Footnote 1 is confusing: I do not understand why the paper mentions model-based RL, and I do not understand what "the dynamics is usually assumed" means.

3. Eq. 3 is a constrained optimization problem, and Eq. 4 is a KL divergence, I do not see why the paper claims that "Eq. 3 can be expressed as Eq. 4".

4. $\pi$ is never defined. Also, $\Pi$ function class is never defined.

5. In Eq. 6, why would taking $\lambda=1$ results in the optimal policy? It seems arbitrary to me unless I missed some important derivation proving that $\lambda=1$ is the optimal Lagrange multiplier.

Also, the theory results in section 4 seem rather trivial, instead, the paper may be improved by showing why these observations are significant.

**Questions:**

See above.

---

> ### Author Response · Authors · 2023-11-20
>
> ### Weakness
> > 1. *The CML and RLM are still different problems. The paper should not claim that they analysis CLM by RLM because they are "intrinsically analogous".*
>
> ---
>
> - Dear Reviewer #2, thank you for your thoughtful feedback. As we mention in *“MR1) of the Meta responses to All Reviewers,”* we initially defined the scope of our work as research on controllable language models (CLMs) because we believed that our work offered an original perspective on CLMs, i.e., controlling LMs can be viewed from a bi-objective perspective, and that this perspective contributes to the field of language model control research in general. However, your comment made us realize that this generalization could be misleading. We have therefore removed the sentence that includes the expression “*intrinsically analogous*“. We thank you for your rigorous comments.
>
> ---
>
> > 2. *Footnote 1 is confusing: I do not understand why the paper mentions model-based RL, and I do not understand what "the dynamics is usually assumed" means.*
>
> ---
>
> - Dear Reviewer #2, thank you for your thoughtful feedback.
> - The reinforcement learning (RL) framework consists of three elements: 1) an agent that learns a policy for making decisions, 2) a reward that is given as a result of the agent's decisions, and 3) an environment model that introduces uncertainty in the environment from which the agent learns.
> - The presence or absence of an environmental model divides the RL category into model-based RL and model-free RL. we assumed that when thinking of reinforcement learning, researchers typically consider the existence of an environment, i.e., model-based RL.
> - Accordingly, given that the primary audience for this paper would be in the NLP Society, we thought it would be more helpful to explicitly mention model-based RL and model-free RL, and then make it clear that RLM belongs to the latter category.
> - However, your feedback has made us realize that mentioning this could have been more confusing. Accordingly, we decided to remove the footnote 1 based on your feedback. Thank you again.
>
> ---
>
> > 3. *Eq. 3 is a constrained optimization problem, and Eq. 4 is a KL divergence, I do not see why the paper claims that "Eq. 3 can be expressed as Eq. 4".*
>
> ---
>
> - Dear Reviewer #2, thank you for your thoughtful feedback.
> - If you look carefully, you'll notice that Eq (3) does not represent a constrained optimization problem; it's just a mathematical representation of RLM as one of many ways to model $\ln p\_{\text{clm}}(\tau|c)$ (just like Eq (1) and Eq (2)).
> - In other words, it's just a mathematical way of saying "RLM is a fine-tuning approach that updates the parameters $\theta\_{\text{lm}}$ so that the pre-trained language model $p\_{\text{lm}}$ is maximized for a reward function $R(\tau)$".
> - Perhaps the notation $s.t.$ is causing some misunderstanding, so I'll change it to notation $\text{where}$.
>     - $\text{Eq (3)} \qquad \ln p\_{\text{clm}}(\tau|c) \approx \ln p\_{\text{lm}}(\tau|\theta\_{\text{clm}}) \qquad          \text{where} \qquad \theta\_{\text{clm}} = \argmax\_{\theta\_{\text{lm}}} \mathbb{E}\_{\tau \sim p\_{\text{lm}}(\hat{\tau}|\theta\_{\text{lm}})} \left[ R(\tau) \right] \ .$
> - Meanwhile, Eq (4) only defines the representative objective function of RLM. Therefore, the claim *“Eq. (3) can be expressed as Eq. (4)”* is simply a statement to explain that Eq (4) is a more refined mathematical fomulation of Eq (3).
>     - $\text{Eq (4)} \qquad \text{KL}\left[ \pi(\tau) \big|\big| \beta(\tau)e^{ R(\tau) } \right] = \sum\_{\tau} \pi(\tau) \ln \frac{ \pi(\tau) }{ \beta(\tau) e^{ R(\tau) } } = - \ \mathbb{E}\_{\tau \sim \pi}[R(\tau)] + \text{KL}[\pi(\tau)||\beta(\tau)]$
>     - Representative objective function of RLM : $\mathbb{E}\_{\tau \sim \pi}[R(\tau)] - \text{KL}[\pi(\tau)||\beta(\tau)]$
>
> - For evidence of the use of Eq (4) as a representative objective function in RLM, please refer to the following references:
>     - [Stiennon et al. (2020)](https://proceedings.neurips.cc/paper/2020/file/1f89885d556929e98d3ef9b86448f951-Paper.pdf), page 6, Section 3.4, Human Feedback Policies paragraph
>     - [Korbak et al. (2022)](https://aclanthology.org/2022.findings-emnlp.77.pdf), Section 3, Equation (4)
>     - [Perez et al. (2022)](https://arxiv.org/pdf/2202.03286.pdf), Section 2.2, Reinforcement Learning paragraph
>     - [Korbak et al. (2022)](https://proceedings.neurips.cc/paper_files/paper/2022/file/67496dfa96afddab795530cc7c69b57a-Paper-Conference.pdf), page 3, Equation (2)
>     - [Ouyang et al. (2022)](https://proceedings.neurips.cc/paper_files/paper/2022/file/b1efde53be364a73914f58805a001731-Paper-Conference.pdf)’s [Supplementary Material](https://proceedings.neurips.cc/paper_files/paper/2022/file/b1efde53be364a73914f58805a001731-Supplemental-Conference.pdf), page 31, Section D.4, Equation (2)
>     - [Bai et al. (2022)](https://arxiv.org/pdf/2204.05862.pdf), page 17, Equation (4.1)

---

> ### Author Response · Authors · 2023-11-20
>
> > 4. $\pi$ is never defined. Also, $\Pi$ function class is never defined.*
>
>  - Dear Reviewer #2, thank you for your thoughtful feedback. But I'm sorry that I don't understand in what context you pointed out that $\pi$ and $\Pi$ are not defined. Is Section 3.2, i.e., the Terms and Notation section, insufficient to provide enough information about the definition of $\pi$? If so, I would appreciate some clarity on this feedback.
>
> ---
>
> > 5. *In Eq. 6, why would taking $\lambda=1$ results in the optimal policy? It seems arbitrary to me unless I missed some important derivation proving that $\lambda=1$ is the optimal Lagrange multiplier.*
>
> - Dear Reviewer #2, thank you for your thoughtful feedback. I understand what your point is. However, my explicit choice of $\lambda = 1$ in Eq (6) (a seemingly arbitrary choice) was intentional, and not arbitrary.
>
>     - $\text{Eq (6)} \qquad \pi^{\*}(\tau) = \frac{\beta(\tau)e^{R(\tau)}}{e^{1-\lambda}} = \beta(\tau)e^{R(\tau)} \quad \left( \lambda=1 \right)$
>
> - To understand that $\lambda=1$ is an optimal choice, not an arbitrary choice, we first need to know that from a probabilistic perspective, the optimal policy $\pi^{\*}$ is always defined as the Boltzmann distribution $\frac{\beta(\tau)e^{R(\tau)}}{\sum\_{\tau}\beta(\tau)e^{R(\tau)}}$ with regardless of $\lambda$.
> - Specifically, given that Eq (6) is the solution of Eq (5) and that solving Eq (5) is equivalent to minimizing Eq (4), we can derive some properties underlying Eq (6) from the perspective of minimizing Eq (4).
>     - $\text{Eq (4)} \qquad \text{KL}\left[ \pi(\tau) || \beta(\tau) e^{R(\tau)} \right] = \sum\_{\tau} \pi(\tau) \ln\frac{\pi(\tau)}{\beta(\tau)e^{R(\tau)}}$
>     - $\text{Eq (5)} \quad \argmax\_{\pi} \mathbb{E}\_{\tau\ \sim \pi} \left[ R(\tau) + \ln \beta(\tau) \right] + \mathcal{H}[\pi] \quad s.t. \quad \sum\_{\tau}\pi(\tau)=1$
>
> - We know that minimizing Eq (4) always results in the same optimal policy $\pi^{\*}$ regardless of $\lambda$, because the denominator becomes a normalization constant due to the probability condition $\sum\_{\tau}\pi(\tau)=1$ (see Eq (10) in Appendix A.1):
>     - Eq (10)    $\pi^{\*}(\tau)=\frac{\beta(\tau)e^{R(\tau)}}{e^{1-\lambda}} =
>             \frac{\beta(\tau)e^{R(\tau)}}{\sum\_{\tau}\beta(\tau)e^{R(\tau)}}  \qquad \left(\ \because \sum\_{\tau}\pi(\tau)=1 \right)$
>
> - We also know that Eq (4) is the KL divergence, a metric that quantifies the distance between two probability distributions, where the minimum value is always equal to 0.
>
> - Accordingly, the condition $\sum\_{\tau} \beta(\tau)e^{R(\tau)}=1$ is necessarily required for Eq (4) to be 0.
>     - $\text{Eq (4)} \qquad \text{KL}[\pi(\tau)||\beta(\tau)e^{R(\tau)}] = -\sum\_{\tau} \pi(\tau)\ln \frac{\beta(\tau)e^{R(\tau)}}{\pi(\tau)} \geq -\ln \sum\_{\tau} \pi(\tau) \frac{\beta(\tau)e^{R(\tau)}}{\pi(\tau)} = - \ln \sum\_{\tau} \beta(\tau)e^{R(\tau)}=0 \quad \bigg( \ \text{if} \ \sum\_{\tau} \beta(\tau)e^{R(\tau)}=1 \bigg)$
>
> - That is,  $\sum\_{\tau} \beta(\tau)e^{R(\tau)}=1$ is the necessary condition for minimizing Eq (4), i.e., solving Eq (5), and therefore this condition must be considered in Eq (10).
>
> - Let us plug $\sum\_{\tau} \beta(\tau)e^{R(\tau)}=1$ into Eq (10). As a result, we can obtain $\pi^{\*}(\tau) = \frac{\beta(\tau)e^{R(\tau)}}{e^{1-\lambda}} = \frac{\beta(\tau)e^{R(\tau)}}{1}$, which implies $e^{1-\lambda}=1 \Leftrightarrow \lambda =1$  must hold.
>
> - In conclusion, given the condition $\sum\_{\tau} \beta(\tau)e^{R(\tau)}=1$ that guarantees the minimum value of Eq (4), it is obvious $\lambda=1$ is the optimal Lagrange multiplier and thus $\pi^{\*} = \beta(\tau)e^{R(\tau)}$ will always hold.

---

> ### Author Response · Authors · 2023-11-22
>
> Dear Reviewer EE6f,
>
> Thank you for your detailed review.
>
> Your comment *"The CML and RLM are still different problems"* helped us clarify the scope of our work, and your comment *"Footnote 1 is confusing"* made us realize that presenting too much information can be confusing from a reader's perspective. Also, your comment *"Why would taking $\lambda=1$ result in the optimal policy?"* reminded us to be more mathematically rigorous about things that we take for granted.
>
> We hope our responses above, including Meta Responses (MR1, MR2, and MR3), answered all of your questions.
>
> If you have any further questions about our answers or the study itself, please let us know.

---

### Official Review · Reviewer_pDqw · 2023-11-03

**Soundness:** 3 good
**Presentation:** 2 fair
**Contribution:** 3 good
**Rating:** 5
**Confidence:** 2

**Summary:**

GOAL: Study Controlled Language Generation Models (CLM) through the lens of RL

OVERALL/The Main Point:

The main point of this work is to point out that CLM is basically a biobjective RL problem. They then show that this insight helps by using an RL trick in the CLM context. The trick does improve results (compared to treating it as a simple RL method). It is unclear whether reframing as an RL problem helps compared to these two baselines:

 1) other methods finetune the LLM per control code but do not frame the problem in an RL manner.

 2) The same RL trick used in this paper used to update the policy, vs only using the trick at controlled decoding time (and no finetuning involved). This baseline is needed since the RL trick used in this paper actually also exists in in other controlled decoding papers: https://aclanthology.org/2022.naacl-main.57.pdf)




OTHER CONTRIBUTIONS:
1. Frame 3 variants on CLM problems as off policy RL that has to max both likelihood of behavior policy and reward score of reward model. This means we can minimize the KL divergence of the target policy and the behavior policy + exponentiated reward. (I’m not clear to me why the reward is exponentiated, other than for math convenience in Eqs 5 and 6)
2. Frame off policy RL as a bi-objective function with a necessary Pareto frontier.
    - Discuss tT=heoretical outcomes of 2 and empirical justification of some properties in a simple 10-turn position game.
3. Reward Dropout is an RL trick that keeps only the top %ile of rewards to guarantee improvement based on Pareto improvement condition. They show that using this RL trick helps controlled generation as motivation for why an RL framing of the CLM problem is useful.

**Strengths:**

1. The proofs and experimental results in Sections 4 and 5 are well written and easy to read within each section. The connection between them is a weakness (see below)

2. Reward dropout is a very simple and effective strategy used to enhance decode-time fine-tuning.

3. CLMs are a common biobjective function, and making the connection to RL is neat.

**Weaknesses:**

1. The experimental results in Section 5 does not provide surprising or insightful results. Even the translation to LLM concepts doesn’t provide insight. The takeaways are that we do better when the two rewards have more overlap (so have a wide span of vocabulary and use an LLM over a smaller model) and when the two rewards have similar maxes (so line up your two rewards by training on data that fulfills both objectives). The visualization is cute, but the takeaways are already standard knowledge (bigger models do better, OOD tasks are harder). If the intent is to empirically test your proofs, then can you write this section in terms of when a policy  becomes a uniform policy (4.2), or show the pareto frontier across policies with the same reward etc. etc. The connection between 4 and 5 is weak.

2. [PRESENTATION]  I’d put a minimal version of Appendix D.2 in the main text -- enough to understand the set-up without reading the full Appendix.

3. Also this set-up described in Appendix D.2 doesn’t really allow you to modify the first part of β(τ ), and I feel that to keep with the spirit of this being a bi-objective RL problem, it should be possible to get different rewards from that first part of β(τ ).

4. [PRESENTATION]  The decoding method is quite expensive, as it involves updating the params of the entire LLM for each target objective (finetuning to be polite is one entire fine-tuning, finetuning to be a negative sentiment is one entire fine-tuning). A lot of CLM methods have the same model be able to output language under different control codes. Can you call this out more explicitly in the limitations section (because it doesn't come across until you read the Appendix)

5. [SOUNDNESS] For the NLP experiment, I’d like to see a non RL-motivated baseline method. The random dropout isn’t really a baseline. It’s nice that this trick from RL translates nicely here, but does the RL framing allow you do better than other CLM methods trained on those datasets? (ex: Perhaps FUDGE as another classifier guided CLM method, or better yet, Diffusion LM adapted to this task perhaps.).

6. [SOUNDNESS/NOVELTY] The last section (the NLP experiment) is strikingly similar to: https://aclanthology.org/2022.naacl-main.57.pdf I'm actually not sure there are any differences except that they don't then re-train the policy (please highlight the other differences for me if they exist). Do you do better than this paper (them only using the method as a controlled decoding solution, and your version fine-tuning based on the controlled decodes)? (A positive answer to this question would also help motivate your RL framing where the policy is typically updated).

7. [NOVELTY] Framing Controlled Decoding as an RL problems has been implicitly done before: https://arxiv.org/abs/1909.09492

**Questions:**

1. In Eq 4, why is the reward function exponentiated? Why does this make sense in context of the problem (and not only because it leads to nice cancellations in successive equations).

2. How is R computed for each sentence?

3. Are there other RL tricks you believe would be useful? Adding those here (and beating CLM baselines AND comparing the effect of updating policy with the trick vs only controlled decoding with the trick) would better help make the case for the utility of the RL-perspective empirically. As is, I'm not convinced this framing is marginally more empirically useful.

4. My own main weakness as a reviewer is that I may be undervaluing the theoretical contributions here. Other works have already referred to controlled decoding as an RL problem, but they do not have your proofs. Either discuss why the theory alone is a solid contribution  (Why is treating CLM as an biobjective RL problem -- as past papers do --  is an unsafe assumption without these proofs established) OR make the case that the RL perspective is useful empirically (See Question 3)

**Details Of Ethics Concerns:**

It's a standard controlled generation use case, that suffers from no ethics concerns that don't apply to all generative text models.

---

> ### Author Response · Authors · 2023-11-20
>
> ### Weakness
> >1. *The experimental results in Section 5 does not provide surprising or insightful results. Even the translation to LLM concepts doesn’t provide insight. The takeaways are that we do better when the two rewards have more overlap (so have a wide span of vocabulary and use an LLM over a smaller model) and when the two rewards have similar maxes (so line up your two rewards by training on data that fulfills both objectives). The visualization is cute, but the takeaways are already standard knowledge (bigger models do better, OOD tasks are harder). If the intent is to empirically test your proofs, then can you write this section in terms of when a policy becomes a uniform policy (4.2), or show the pareto frontier across policies with the same reward etc. etc. The connection between 4 and 5 is weak.*
>
> - Dear Reviewer #1, thank you for your thoughtful feedback. Your feedback, especially *"Section 5 does not provide surprising or insightful results"* and *"the takeaways are already standard knowledge"*, was very thought-provoking for us. We have had a lot of fun preparing this response and would like to express our sincere gratitude for giving us the opportunity to respond.
> - We believe that the goal of Section 5 is best captured by the last sentence of the first paragraph in Section 5: *"The goal of this experiment is to confirm the theoretical results and analyze them in the CLM context."* That is, we wrote Section 5 to support the validity of the theoretical outcomes presented in Section 4, including theorems and propositions.
> - The most important contents of our work are Theorems 4.1, 4.2, and 4.3 and the simple yet powerful technique called Reward Dropout that derives from them. **Therefore, I sincerely request that you read *“MR2) of the Meta Responses”* first.**
> ---
> - As explained in *“MR2) of the Meta Responses”*, **Reward Dropout is built on Theorems 4.1, 4.2, and 4.3**, and thus we need to verify how reliable these theorems are before examining the effectiveness of Reward Dropout. More precisely, we need to verify whether our theorems can derive *“obvious facts”*, *"trivial properties"* or, to use your words, *"unsurprising takeaways."* **If our theorems fit well with obvious facts** (known as *"standard knowledge"*) **without contradiction,** then we can say that the theorems are reliable, and by extension, **we can say that the theoretical background for Reward Dropout is reliable as well.**
> - **Accordingly, we have chosen two specific questions** that can be applied to these theorems. **Our aim is to find answers to these questions and ensure that these answers are aligned with *“standard knowledge”***. The two questions we have selected are as follows:
>     - *“Can behavioral policy improve the Pareto optimum of target policy (auxiliary condition; Proposition 4.1)?”*
>     - *“What is the Pareto optimum of the target policy if the behavioral policy does not have a feasible region (violation condition; Proposition 4.2)?”*
>     - The reasoning behind the above questions is straightforward. We consider RLM as a bi-objective optimization problem, and it is important for optimization research to understand the conditions under which a better local optimum can be obtained (auxiliary conditions) and the conditions under which a solution cannot exist (violation conditions).
>
> - **Proposition 4.1 follows from Theorem 4.2 and states that the expected reward of the optimal target policy $\mathbb{E}\_{\tau^{} \sim \pi^{}}[R(\tau)]$ can be improved**. This provides the answer to *“Can behavioral policies improve the Pareto optimum of a target policy?”,* which implies that *“the larger the capability of the behavioral policy, the better the expected reward of the optimal target policy.”* **This aligns well with our standard knowledge *"bigger models do better"*** without contradiction.
>     - $\text{(Theorem 4.2)} \qquad  \mathbb{E}\_{\tau^{*} \sim \pi^{\*}} [ R(\tau) ] = - \mathbb{E}\_{\tau^{\*} \sim \pi^{\*}} [ \ln \beta(\tau)]$
>     - $\text{(Proposition 4.1)} \qquad \mathbb{E}\_{\tau^{*} \sim \pi^{\*}} [ R(\tau) ] = - \mathbb{E}\_{\tau^{\*} \sim \pi^{\*}} [ \ln \beta(\tau)] \quad \Longrightarrow \quad \mathbb{E}\_{\tau^{\*} \sim \pi^{\*}}[R(\tau)] \propto \mathcal{H}[\beta(\tau)]$

---

> ### Author Response · Authors · 2023-11-20
>
> ### (cont.)
> >1. *The experimental results in Section 5 does not provide surprising or insightful results. Even the translation to LLM concepts doesn’t provide insight. The takeaways are that we do better when the two rewards have more overlap (so have a wide span of vocabulary and use an LLM over a smaller model) and when the two rewards have similar maxes (so line up your two rewards by training on data that fulfills both objectives). The visualization is cute, but the takeaways are already standard knowledge (bigger models do better, OOD tasks are harder). If the intent is to empirically test your proofs, then can you write this section in terms of when a policy becomes a uniform policy (4.2), or show the pareto frontier across policies with the same reward etc. etc. The connection between 4 and 5 is weak.*
>
> - **Proposition 4.2 follows from Theorem 4.1 and states that if the behavioral policy is ill-defined for all target trajectories $\forall{\tau} \sim \pi, \ \ \beta(\tau)=0$,** (i.e., no solution exists because the behavioral policy cannot access a feasible region), then the RUBO disappears, and the bounded RL problem represented by Eq (4) or Eq (5) is transformed into an unbounded entropic RL problem where the original reward plus entropic reward are maximized together. This provides the answer for *“What is the Pareto optimum of the target policy if the behavioral policy does not have a feasible region.”* As a result, the target policy will converge to a uniform policy with the highest entropy, i.e., a random policy. **Therefore, we should collect as much data as possible so that the pre-training dataset of the behavioral policy contains feasible regions, i.e., the pre-training dataset of the behavior policy is well-defined, for the control objective (reward objective).** This aligns well with our standard knowledge **"*more data do better*"** without contradiction.
>     - $\text{(Theorem 4.1)} \qquad \mathbb{E}\_{\tau \sim \pi} \left[ R(\tau) \right] \leq \text{KL} \left[ \pi(\tau) || \beta(\tau) \right] = \text{RUBO}$
>     - $\text{(Proposition 4.2)} \qquad \mathbb{E}\_{\tau \sim \pi} \left[ R(\tau) \right] \leq \text{KL} \left[ \pi(\tau) || \beta(\tau) \right] \ \overset{\forall{\tau} \sim \pi, \ \ \beta(\tau)=0}{{\Longrightarrow}} \ \ \mathbb{E}\_{\tau \sim \pi}[R(\tau)] + \mathcal{H}[\pi] \leq +\infty$
>     - To emphasize the implications of Proposition 4.2, we have provided the following example in the text: *“For example, let us say we need to control the target LM so that sentences are generated with negative sentiment. If the behavior LM is pre-trained on a dataset consisting only of positive sentences (i.e., the behavior policy is ill-defined), then the behavior LM cannot provide negative candidate sentences.“* Please refer to the text highlighted in orange in section 5 of the revised submission.
> ---
> - **In Section 5, we confirmed that Theorems align well with standard knowledge** (e.g., *"bigger models do better"*, "*more data do better"*). Fig 2(b) illustrates this well. The first column confirms the statement of Proposition 4.1 and the second column confirms the statement of Proposition 4.2. With these results, we can consider the theorems, and the theoretical background of Reward Dropout built on top of them, to be sufficiently reliable. With this reliability, the effectiveness of reward dropout was further validated by Fig 2.
> ---
> - **We believe that the content of Section 5 will be of great interest to the NLP Society, and in particular to groups working on RLM**, because Eq (4) is the typical RLM objective function, and our theorems and propositions all stem from Eq (4). **Section 5 will convey messages that are essential to modern RLM researchers**, such as the existence of a tradeoff between $\beta(\tau)$ and $R(\tau)$, the possibility of Pareto improvement through reward manipulation, and the importance of large models and large amounts of data.
>     - $\text{(Eq 4)} \qquad \text{KL}\left[ \pi(\tau) || \beta(\tau) e^{R(\tau)} \right] = \sum_{\tau} \pi(\tau) \ln\frac{\pi(\tau)}{\beta(\tau)e^{R(\tau)}}$
>     - $\text{(Objective Function of RLM)} \qquad \mathbb{E}\_{\tau \sim \pi_{\phi}^{\text{RL}}} [R(\tau)] - \alpha\text{KL}[\pi_{\phi}^{\text{RL}}(\tau)||\pi^{\text{SFT}}(\tau)] \quad \overset{\times -1}{\underset{\alpha=1}{\Longrightarrow}} \quad \text{KL}[\pi_{\phi}^{\text{RL}}(\tau)||\pi^{\text{SFT}}(\tau)] - \mathbb{E}\_{\tau \sim \pi_{\phi}^{\text{RL}}}[R(\tau)] = \sum_{\tau} \pi_{\phi}^{\text{RL}}(\tau) \ln \frac{\pi_{\phi}^{\text{RL}}(\tau)}{\pi^{\text{SFT}}(\tau)e^{R(\tau)}} \ \cdots \ \text{Eq (4)}$

---

> ### Author Response · Authors · 2023-11-20
>
> > 2. *[PRESENTATION] I’d put a minimal version of Appendix D.2 in the main text -- enough to understand the set-up without reading the full Appendix.*
>
> - Dear Reviewer #1, thank you for your thoughtful feedback. Initially, Appendix D.2 was involved in the body, but we moved it to the Appendix due to the page limit. I agree with your opinion. I had put a *minimal version of Appendix D.2* in the main text (highlighted the text in orange color). Please refer to the Implementation Details paragraph in Section 6.

---

> ### Author Response · Authors · 2023-11-20
>
> > 3. *Also this set-up described in Appendix D.2 doesn’t really allow you to modify the first part of β(τ ), and I feel that to keep with the spirit of this being a bi-objective RL problem, it should be possible to get different rewards from that first part of β(τ ).*
>
> - Dear Reviewer #1, thank you for your thoughtful feedback. We understand your concerns.
> - However, as the mathematical notation throughout the paper and explicit references in Section 2.2 and Appendix D.1 indicate, it is the trajectory or sentence $\tau$ that we consider as a variable in this study. The reason we followed the mathematical framework of probabilistic inference described in Section 2.2 rather than the traditional RL framework is that we wanted to treat *“the entire trajectory as a random variable,”* not just the actions, states, or parts of the trajectory.
> - In this sense, fixing the initial part of $\tau$ is not a problem. Of course, we agree that the fact that the initial part of $\tau$, defined as a random variable, depends on a fixed prefix, i.e., *“a constant variable,”* may fade the meaning of *"random variable"*. However, what is important to us is the complete sentence (complete trajectory) sampled starting from the prefix, and since there are theoretically an infinite number of possible complete sentences, I don't see a big problem with interpreting $\tau$ as a random variable.
> - To summarize,
>     1. Eq (6), which formalizes the bi-objective spirit, originates from the "probabilistic inference" problem of minimizing Eq (4).
>     2. And the random variable defined in this paper is the complete trajectory $\tau$, not a part of it.
>     3. Therefore, if there is no problem with interpreting $\tau$ as a random variable as a complete sentence, then fixing a part of $\tau$ does not go against the spirit of the bi-objective

---

> ### Author Response · Authors · 2023-11-20
>
> > 4. *[PRESENTATION] The decoding method is quite expensive, as it involves updating the params of the entire LLM for each target objective (finetuning to be polite is one entire fine-tuning, finetuning to be a negative sentiment is one entire fine-tuning). A lot of CLM methods have the same model be able to output language under different control codes. Can you call this out more explicitly in the limitations section (because it doesn't come across until you read the Appendix)*
>
> - Dear Reviewer #1, thank you for your thoughtful feedback. Your feedback is certainly valid. Unlike other approaches such as CCLM and BCLM that allow *"a single model to generate text from different control codes,"* what you point out is a common problem with RLM-based approaches. We will discuss this in the limitations section as page limits allow.

---

> ### Author Response · Authors · 2023-11-20
>
> > 5. *[SOUNDNESS] For the NLP experiment, I’d like to see a non RL-motivated baseline method. The random dropout isn’t really a baseline. It’s nice that this trick from RL translates nicely here, but does the RL framing allow you do better than other CLM methods trained on those datasets? (ex: Perhaps FUDGE as another classifier guided CLM method, or better yet, Diffusion LM adapted to this task perhaps.).*
>
> ---
>
> - Dear Reviewer #1, thank you for your thoughtful feedback.
> - As we mention in *“MR1) of the Meta responses to All Reviewers,”* we initially defined the scope of our work as research on controllable language models (CLMs) because we believed that our work offered an original perspective on CLMs, i.e., controlling LMs can be viewed from a bi-objective perspective, and that this perspective could contribute to the field of language model control research in general. However, we believe that this generalization may be misleading. Therefore, **we clarify the scope of our research as *"Reinforced Language Models (RLMs) that control the generation of language models (LMs) through reinforcement learning (RL)."***
> - We reiterate that the methodological contribution of this study is *"the finding that language models based on a reinforcement learning framework can learn better through a technique called reward dropout, in which a portion of the observed reward is intentionally dropped when utilizing RLM as a control approach.”*
> - Meanwhile, even if we limit the scope of this study to RLMs, we agree that meaningful insights can be obtained by comparing the performance of RLMs with reward dropouts to non-RL-based CLMs belonging to BCLM and CCLM. Therefore, we will conduct a comparative evaluation using FUDGE and CTRL as the baseline, which belong to BCLM and CCLM, respectively. The comparative evaluation is currently delayed due to computing resource issues in the lab. We will let you know the results of the evaluation as soon as the problem is resolved.

---

> ### Author Response · Authors · 2023-11-20
>
> > 6. *[SOUNDNESS/NOVELTY] The last section (the NLP experiment) is strikingly similar to: [https://aclanthology.org/2022.naacl-main.57.pdf](https://aclanthology.org/2022.naacl-main.57.pdf) I'm actually not sure there are any differences except that they don't then re-train the policy (please highlight the other differences for me if they exist). Do you do better than this paper (them only using the method as a controlled decoding solution, and your version fine-tuning based on the controlled decodes)? (A positive answer to this question would also help motivate your RL framing where the policy is typically updated).*
>
> ---
>
> - Dear Reviewer #1, thank you for your thoughtful feedback. we have skimmed the paper you referenced. Unfortunately, we are not sure that your reference and our work are appropriate to compare, for the following reasons:
>     - Our work belongs to the class of RLMs.
>     - The reference work, NeuroLogic A*esque paper, seems to belong to the class of BCLMs.
>
> - Nevertheless, we believe that there are some similarities between the two papers in terms of finding paths, or trajectories, that maximize future (expected) value. Therefore, we will contrast the differences between the two papers to the best of our knowledge, to make the contribution of our work more explicit.
>     - **Our work is a train-time approach**, considering the language model as a policy and optimizing the policy for a control objective.
>         - On the other hand, **the reference work is an inference-time approach,** predicting tokens’ future value subject to the control objective and steering the decoding direction accordingly.
>     - **Our work proposes an RL technique.**
>         - On the other hand, **the reference work proposes a decoding algorithm.**
>     - **The Reward Dropout** proposed in our paper intervenes in the **"training process"** and improves **"training performance"** based on **"theoretical grounds."**
>         - On the other hand, **the NeuroLogic Aesque Decoding** proposed in the reference paper intervenes in the **"inference process"** and improves **"decoding efficiency"** based on **"heuristic grounds.".**

---

> ### Author Response · Authors · 2023-11-20
>
> > 7. *[NOVELTY] Framing Controlled Decoding as an RL problems has been implicitly done before: [https://arxiv.org/abs/1909.09492](https://arxiv.org/abs/1909.09492)*
>
> ---
>
> - Dear Reviewer #1, thank you for your thoughtful feedback. Yes, apart from the paper you just recommended, there have been various studies on controllable generation (including controlled decoding) in the context of RL problems, which we believe demonstrates that this is a very active and vibrant research area.
> - Meanwhile, the novelty of our work is that we do not view controllable generation simply as an RL problem, but rather as a bi-objective problem, and even a Pareto optimization problem. In particular, instead of "assuming" that the likelihood objective and the reward objective are in conflict, we mathematically "prove" that they are in fact in trade-offs. Based on this, we proposed a simple yet powerful technique called Reward Dropout, which we believe is the biggest novelty and contribution of our work.
> - Thank you for recommending a good paper.

---

> ### Author Response · Authors · 2023-11-20
>
> ### Questions
> > 1. *In Eq 4, why is the reward function exponentiated? Why does this make sense in context of the problem (and not only because it leads to nice cancellations in successive equations).*
>
> - Dear Reviewer #1, thank you for your thoughtful feedback. You might have asked this question lightly, but for us, your question had a very big implication. Eq (4) is taken for granted in RLM research as the objective function of RLM, and it was very refreshing for me to see that Eq (4) itself could be questioned. It was also a learning experience for me. We have answered this question in *“MR3) of the Meta responses to All Reviewers.”* Please refer to MR3.
>
> ---
>
> > 2. *How is R computed for each sentence?*
>
> - Dear Reviewer #1, thank you for your thoughtful feedback.  As is usual in RLM research, we pretrained the reward model to compute the rewards. The pre-trained reward model predicts the probability that the generated sentence has the intended attribute. Please refer to the Pseudo algorithm table in Appendix D.4.
>
> ---
>
> > 3. *Are there other RL tricks you believe would be useful? Adding those here (and beating CLM baselines AND comparing the effect of updating policy with the trick vs only controlled decoding with the trick) would better help make the case for the utility of the RL perspective empirically. As is, I'm not convinced this framing is marginally more empirically useful.*
>
> - Dear Reviewer #1, thank you for your feedback. Your question about other RL 'tricks' that we believe useful sounded like a 'tricky' question, and to be honest, it is because we didn't understand what you meant by the term 'RL trick'.
>
> - Of course, there are many “tricky techniques “ that help improve RL performance. For example, adding a baseline to the reward is one of the simplest techniques you can use. This technique fixes the reward near the baseline, making the objective function that optimizes the reward an unbiased estimator, thereby reducing the variance of trajectory sampling and improving the stability of model training.
>
> - Another example of a simple and tricky technique is reward clipping, which is a common technique used in DQNs. Reward clipping was designed to improve the robustness of a model by reducing the impact of extreme values, and is widely used as an alternative to gradient clipping; in fact, Reward Dropout is the philosophical opposite of reward clipping in that it focuses on amplifying the impact of extreme values.
>
> - If the term RL trick you mentioned means the “reward-driven” RL techniques as simple and tricky as the examples above, as far as we know, there are not many such simple & tricky techniques. In particular, most of them focus on training stability, like reward baseline and reward clipping introduced above, and we have not yet seen any techniques that focus on improving control performance, like Reward Dropout. In this sense, Reward Dropout can be said to be very original and novel since it aims to improve control performance while being simple and tricky.
>
> - **we are strongly confident that Reward Dropout will become a very influential technique in the future because it not only definitely improves the performance of the control, but also can be applied to any method, algorithm, or task** as long as it deals with RL problems. Considering the high applicability of the RL framework despite its high computational complexity, **I believe that more and more research will be based on RL in the future, and the importance of Reward Dropout will increase accordingly.**
>
> ---
>
> > 4. *My own main weakness as a reviewer is that I may be undervaluing the theoretical contributions here. Other works have already referred to controlled decoding as an RL problem, but they do not have your proofs. Either discuss why the theory alone is a solid contribution (Why is treating CLM as an biobjective RL problem -- as past papers do -- is an unsafe assumption without these proofs established) OR make the case that the RL perspective is useful empirically (See Question 3)*
>
> - Dear Reviewer #1, thank you for your thoughtful feedback. We think our response to *“weakness #1”*, and *“MR2) and MR3) of the Meta Responses”* might be the answer to this question.

---

> ### Author Response · Authors · 2023-11-22
>
> Dear Reviewer pDqw,
>
> From the volume of your feedback, your sincerity regarding our research came through to us. Your comments have provided us with a great deal of intellectual stimulation.
>
> For example, your comment *"The experimental results in Section 5 does not provide surprising or insightful results."* was very thought-provoking. It is because we thought showing our theorems are linked to *obvious facts* was not only what we intended but also most importantly to support the reliability of our theorems. When we first saw the above comment, we were puzzled, but in preparing the response, we found it very intellectually stimulating. Similarly, the question *"In Eq 4, why is the reward function exponentiated?"* was mind-blowing to us because we never questioned it. It was a refreshing experience for us.
>
> Also, your question *"Are there other RL tricks you believe would be useful?"* made us realize that mentioning some well-known reward-driven tricks and comparing them to the proposed technique, Reward Dropout, would be a very intuitive guide for the readers. We think this comment will help improve the quality of our paper for sure. Your comments related to Appendix D.2, e.g., *" I’d put a minimal version of Appendix D.2 in the main text"* and *"Also this set-up described in Appendix D.2 doesn’t really allow you to modify the first part of $\beta(\tau)$"*,  were also very helpful in improving the sophistication of our work.
>
> Once again, thank you for your detailed and constructive feedback.
>
> In response to your feedback, and in terms of reconsidering the value of our study, we would like to ask you to kindly read our MR1, MR2, and MR3.
>
> If you have any additional comments, please do not hesitate to let us know.

---

> ### Comment · Reviewer_pDqw · 2023-11-22
>
> Your contribution is to show: language models based on a reinforcement learning framework can learn better through a technique called reward dropout...
>
> My point here is that the use of your specific reinforcement learning framework for such models also needs some motivation. Restated, something must be gained by using this new perspective: better results or some brand new method.
>
> If there is a simpler method and framing that comes up with an analogous method (I think it is a strict advantage to not have to do a train time approach which is usually far more costly) and achieves better results, then I don't feel the case for this new framing is made quite as strongly.
>
> Is your claim here that the fact that this paper's method is based on theory, what makes it better than past framings with very similar methods? (That's fine by me. I just want to be sure I'm grasping the claim).

---

> ### Author Response · Authors · 2023-11-23
>
> - Dear Reviewer pDqw, thank you for clarifying your point.
> - First of all, to the best of my knowledge, the advantages and disadvantages of CCLM, BCLM, and RLM can be summarized as follows:
>     - CCLMs (Class Conditional Language Models)
>         - Pros
>             - **Easy to Implement** : All we need is to prepare an LM and prepend a sentence with a prefix for control.
>             - **Low Training Cost** : The training cost is extremely low because CCLMs only need to prepend the sentence with one control token corresponding to the control class and simply train the LM in a supervised learning manner.
>             - **Low Inference Cost** : The inference cost is low because CCLMs only need to prepend one control token corresponding to the control class before the prefix/prompt (e.g., <bos> token), and then generate sentences.
>             - **Easy to Deploy** : Once training-time control is completed, CCLMs are easy to deploy in real-world services
>             - **High Scalability** : CCLM is easy to scale to model with multiple control objectives because, once a multi-class control token vector, rather than a single control token, is prepended before the sentence, then at inference time, we can feed any class of control tokens.
>         - Cons
>             - **Inflexible Control** : Because a CCLM injects discrete signals based on class tokens as inputs to the LM, there is less flexibility to adjust the strength of the control. For example, for sentiment control (negative vs. positive), CCLMs can only control with a polarity that falls into either of two classes: positive or negative, and CCLMs cannot generate sentences that fall on an arbitrary continuum between the two (e.g., a review that is 75% positive + 25% negative).

---

> ### Author Response · Authors · 2023-11-23
>
> - Dear Reviewer pDqw, thank you for clarifying your point.
> - First of all, to the best of my knowledge, the advantages and disadvantages of CCLM, BCLM, and RLM can be summarized as follows (cont.):
>     - BCLMs (Bayesian Controllable Language Models)
>         - Pros
>             - **Easy to Implement** : All we need is to prepare an LM, pre-train the classifier to the control objective, compute the probability of the on-the-fly tokens generated by LM, and sum it with the probability that the $t$-th token is classified into a specific control class.
>             - **Flexible Control** : Because BCLM utilizes a continuous-valued signal from the classifier (i.e., the probability that a sentence belongs to a particular control class), the control strength can be adjusted flexibly. For example, for sentiment control (negative vs. positive), every $t$-th token sampled from LM can be fed into a pre-trained classifier to force it to only sample tokens with a predicted probability of 75% positive + 25% negative.
>             - **Low Training Cost** : The training cost is low because BCLMs only need to pre-train a classifier (that predicts the probability that a sentence belongs to a particular control class).
>             - **High Scalability** : BCLM is easy to scale to model with multiple control objectives because once a multi-class classifier is trained, then we can compute the probabilities of tokens belonging to any classes.
>         - Cons
>             - **High Inference Cost** : The inference cost is extremely high because BCLMs require a Monte Carlo sampling. This Monte Carlo approach is expensive a lot in that sampling each token up to a complete sentence is required for each mini-batch sample. Furthermore, BCLMs are the inference-time approach that computes the probabilities (e.g., how likely the $t$-th token (1) to be sampled and (2) to be classified to a specific control class) of on-the-fly tokens generated by LM, and sum the probabilities to adjust sampling distribution to be biased toward control objective. Therefore, the computation complexity at inference time is extremely high.
>             - **Hard to Deploy** : Since inference-time control is required, BCLMs are difficult to deploy in real-world services.
>     - RLMs (Reinforced Language Models)
>         - Pros
>             - **Easy to Implement** : All we need is to prepare an LM, pre-train the reward model to the control objective, and fine-tune LM with the reward model.
>             - **Flexible Control** : Because RLMs utilize a continuous-valued signal from the reward model (i.e., the probability that a sentence belongs to a particular control class), the control strength can be adjusted flexibly. Furthermore, instead of having to compute the reward for each $t$-th token, RLMs can utilize the reward of an entire sentence (i.e., the entire trajectory). For example, for sentiment control (negative vs. positive), the entire sentence sampled from the LM can be fed into a pre-trained reward model and perform reward maximization only for the sentences with a predicted probability of 75% positive + 25% negative.
>             - **Low Inference Cost** : The inference cost is extremely low because, once trained, RLMs only need to generate sentences from the prefix/prompt (e.g., <bos> token).
>             - **Easy to Deploy** : Once training-time control is completed, RLMs are easy to deploy in real-world services.
>         - Cons
>             - **High Training Cost** : The training cost is extremely high because RLMs need to pre-train both LM and reward model, and then fine-tune (align) LM with the reward model. For fine-turning, RLMs require a Monte Carlo sampling. This Monte Carlo approach is expensive in that sampling each token up to a complete sentence is required for each mini-batch sample.
>             - **Low Scalability** : RLM is difficult to scale to model with multiple control objectives because separate models must be trained for each control class.

---

> ### Author Response · Authors · 2023-11-23
>
> - As described above, **there are clear advantages and disadvantages to each approach**, and we don't believe that **one approach is necessarily superior to the other.**
> - **Nevertheless, here's why we chose to go with RL framing:**
>     - **First, it's simple to implement**. Of course, there may be differences in details such as the design of the algorithm, model, network structure, and training framework, but in essence, RLM is all about preparing separate LM and Reward models and then making them interact with each other under a reinforcement learning framework.
>     - **Second, RLM has a significant advantage at inference time-complexity,** which is probably the most important aspect of deployment performance. RLM requires high time complexity in the training stage, but the time complexity required in the inference stage is no different from the usual autoregressive LM model.
>     - **Third, "controllability" is theoretically guaranteed.** From the "Policy Improvement Theorem", a well-known theorem in reinforcement learning, **it is guaranteed that training under the RL framework always converges to a local optimum**. We don't know if such a theoretical guarantee of controllability exists in the field of CCLM and BCLM, but **at least if we train with RLM, the control performance at the local optimum is guaranteed.**
>         - We think that the theoretical guarantee of controllability is a tremendous advantage if we extend the scope of RLM to the problem of controlling sequences (e.g., driving sequences, stock price sequences, compound sequences, DNA sequences) not just controlling natural language generation.
>         - In a model where controllability is not theoretically guaranteed, there is no certainty that control will succeed, or even that it has succeeded. When applied to technologies such as autonomous driving, such models can lead to major accidents.
>
> ---
>
> - In conclusion
>     - **Reward Dropout is a technique that always guarantees to improve the control performance** of an arbitrary model under the RL framework. This means that **the Improvement achieved by Reward Dropout addresses the training time complexity that we would otherwise have to sacrifice to guarantee controllability**.
>         - To guarantee controllability, we can decide to use RL frameworks in exchange for the training-time complexity.
>         - In Sections 4 and 5, you can see that we start with much higher performance when applying reward dropouts. This means that less time is needed to train the model.
>
>     - As such, the implication of **using Reward Dropout is that controllability and training efficiency** (i.e., improved optimization) **can be achieved together.**
>     - As a result, we believe that **utilizing Reward Dropout can alleviate the complexity of training time in using RLMs, which is a relative disadvantage compared to other approaches** such as CCLM or BCLM.
>
> ---
> - We have incorporated the aforementioned points into the revised Section 7, "Limitations & Concluding Remarks."
> - We hope our responses and revisions help address your questions and concerns. Thank you once again for your thoughtful and constructive feedback!

---

### Author Response · Authors · 2023-11-20

# Meta Responses to All Reviewers.

---

- To the reviewers, I want to say thanks for your kind and thoughtful feedback. Your feedback helped us make our research more specific and clear about what it contributes and what its limits are.

- Before responding to each reviewer’s feedback individually, we take this space to share the key questions raised by some reviewers and provide our three Meta Responses (MR1, MR2, MR3) to them, as a way of clarifying the scope and problem statement of our research and emphasizing the theoretical and methodological contributions.

- **Meta responses (especially, MR2 and M3) will help everyone better understand the whole picture of our research. So, please check them out.**

---
- We updated our submission to include new experiments that better demonstrate the value of our work. Please check the following:
    - Appendix I
    - Appendix J
    - We moved content that was originally in Appendices I and J into the main body (see Sections 4 and 5).

---
- We will craft and update our submission as soon as possible, at least before the final decision notification. The expected revision will include:
    - We will clarify the research scope.
        - In this regard, we changed our title to  *"Reward Dropout Improves Control: Bi-objective Perspective on Reinforced LM"*
        - To refocus the scope of our paper on RLM, we have removed or reduced remarks related to BCLM or CCLM.
    - We strengthened the implication of Section 4 and the connectivity between Section 3 and Section 4. Please read MR2) of Meta Responses to All Reviewers.
    - We changed the order of the theorems to clarify the theoretical contribution of our work.
        - Theorem 4.2 -> Theorem 4.3
        - Theorem 4.3 -> Theorem 4.2
    - We moved Section 5 (10-turn positioning part) and its related materials, i.e., Propositions 4.1 and 4.2, to  Appendix H.
    - Instead, we moved content that was originally in Appendices I and J into the main body (see Sections 4 and 5).
    - We added a Limitation section at the end of the paper, which describes the shortcomings of the RLM approach compared to other CLM approaches such as CCLM and BCLM.


---
 - For readability, we systematically refer to individual reviewers by number rather than anonymized English codes, as shown below:
    - Reviewer #1 = Reviewer pDqw
    - Reviewer #2 = Reviewer EE6f
    - Reviewer #3 = Reviewer dnj6

---

> ### Author Response · Authors · 2023-11-20
>
> # MR1) Research Scope of Our Work
>
> ---
>
> - We initially defined the scope of our research as the study of controllable language models (CLMs) because we believed that our work offers an original perspective on CLMs, i.e., controlling LMs can be viewed from a bi-objective perspective, and that this perspective contributes to the field of language model control research in general. However, feedback from reviewers has made us realize that this generalization may be misleading. **Therefore, we clarify the scope of our research as *"Reinforced Language Models (RLMs) that control the generation of language models (LMs) through reinforcement learning (RL)."*** With this clarification and narrowing of scope, we can summarize the key takeaways of our paper as follows:
>
>     1. **(Sections 3 and 4)** **We theoretically analyze the objective function of an RLM represented by Eq. (4) from the bi-objective perspective.** In particular, unlike the existing RLM literature that simply focuses on maximizing both objectives (i.e., the likelihood objective $\beta(\tau)$ and the reward objective $R(\tau)$) simultaneously, we focus on the trade-off relationship between the two objective functions, i.e., training an RLM is a Pareto Optimization problem, and we show that there are fundamental limitations to maximizing both objective functions simultaneously.
>
>     2.  **(Section 5) Empirical analysis is performed to support the validity of theoretical outcomes.** Specifically, we check whether some facts about LLM control, which are taken for granted based on intuition or experience, are correctly reproduced on the theoretical foundation we have presented. By demonstrating these facts under deductive logic, we can reaffirm their implications for LLM control.
>
>     3.  **(Sections 4 and 6)** **Finally, we propose a simple, yet powerful technique called Reward Dropout.** The idea behind Reward Dropout stems from Theorems 4.1, 4.2, and 4.3 in Section 4.
>
> -  Reviewer #1, Reviewer #2 and Review #3 have raised some questions related to this response. We sincerely appreciate the reviewers for their thoughtful feedback.

---

> ### Author Response · Authors · 2023-11-20
>
> # MR2) Theoretical and Methodological Contributions
>
> ---
>
> - A key value of our work is that the proposed method, namely Reward Dropout, is built on a theoretical foundation of bi-objective optimization. In other words, our theoretical and methodological contributions are very tightly and seamlessly connected. However, we regret that we have not sufficiently conveyed the connection between the theoretical foundation and the methodology. Therefore, here we briefly describe the main theoretical results presented in Section 4, namely Theorems 4.1, 4.2, and 4.3, and summarize how they led us to propose Reward Dropout.
> ---
> 1. **Theorem 4.1 shows that** the objective function of the RLM represented by Eq (4) implies a trade-off between the two objectives $R(\tau)$ and $\beta(\tau)$. This is confirmed by Reward Upper BOund (RUBO), i.e.,  $\mathbb{E}\_{\tau \sim \pi} \left[ R(\tau) \right] \leq \text{KL} \left[ \pi(\tau) || \beta(\tau) \right]$, which is obtained by arranging Eq (4). It is noteworthy that **RUBO provides us with an interesting intuition: the upper bound of reward is defined by $\text{KL}\left[ \pi(\tau) \big|\big| \beta(\tau) \right]$,** and this implies that *“the larger the KL divergence between $\pi(\tau)$ and $\beta(\tau)$, the higher the expected reward $\mathbb{E}\_{\tau\sim\pi}[R(\tau)]$.”* That is, for $\tau \sim \pi$, the increase in the reward objective $R(\tau)$ above the RUBO results in the decrease in likelihood objective $\beta(\tau)$ due to $\pi$’s divergence from $\beta$. Reversely, the increase in $\beta(\tau)$, i.e.,  $\pi$’s convergence to $\beta$ (= the lower $\text{KL}[\pi(\tau)||\beta(\tau)]$), leads to the decrease in $R(\tau)$. This trade-off relation between $R(\tau)$ and $\beta(\tau)$ suggests that the RLM objective represented by Eq (4) is a bi-objective problem.
>     - $\text{Eq (4)} \qquad \text{KL}\left[ \pi(\tau) || \beta(\tau) e^{R(\tau)} \right] = \sum_{\tau} \pi(\tau) \ln\frac{\pi(\tau)}{\beta(\tau)e^{R(\tau)}}$
>     - $\text{(Theorem 4.1)} \qquad \text{KL}\left[ \pi(\tau) || \beta(\tau) e^{R(\tau)} \right] \Longrightarrow \mathbb{E}\_{\tau \sim \pi} \left[ R(\tau) \right] \leq \text{KL} \left[ \pi(\tau) || \beta(\tau) \right] = \text{RUBO}$
> ---
> 2. This bi-objective nature of the RLM objective function is also revealed in Eq (6), which describes the optimal policy $\pi^{\*}$ obtained by solving Eq (5) or obtained by minimizing Eq (4). **However, Eq (6) only shows that the trajectory from the optimal policy $\tau^{\*} \sim \pi^{\*}$ must have high values in $\beta(\tau)$ and $e^{R(\tau)}$ simultaneously, but it does not tell us $\beta(\tau)$ and $R(\tau)$ are “in trade-offs.” In this sense, Theorem 4.1 is insightful in itself**, in that it presents the existence of a trade-off between $\beta(\tau)$ and $R(\tau)$. To the best of our knowledge, **the explicit discussion on the existence of the trade-offs between likelihood and reward objectives has not received attention in existing RLM studies** that train RLM models using Eq. (4) as the objective function, as we do. In particular, we have never seen cases where the existence of trade-offs has been deductively derived from Eq. (4). **Accordingly, we believe that this is an original and novel contribution of our work.**
>     - $\text{Eq (5)} \qquad \argmax_{\pi} \mathbb{E}\_{\tau\ \sim \pi} \left[ R(\tau) + \ln \beta(\tau) \right] + \mathcal{H}[\pi] \quad s.t. \quad \sum_{\tau}\pi(\tau)=1$
>     - $\text{Eq (6)} \qquad \pi^{\*}(\tau)=\frac{\beta(\tau)e^{R(\tau)}}{e^{1-\lambda}} = \beta(\tau)e^{R(\tau)} \quad \left( \because \lambda = 1 \right)$
>     - As for why it should be $\lambda=1$, please refer to our response to Weakness 5 from reviewer #2.

---

> ### Author Response · Authors · 2023-11-20
>
> # MR2) Theoretical and Methodological Contributions (cont.)
>
> ---
>
> 3. **Theorem 4.2 shows that** as long as RUBO holds, we can verify that **$\beta(\tau)$ and $R(\tau)$ have a negative logarithmic relationship for all optimal solutions, i.e., $\forall \tau^{\*} \sim \pi^{\*}$** where $\pi^{\*}$ is given by Eq (6), i.e., $\pi^{\*}(\tau) = \beta(\tau)e^{R(\tau)}$. **This suggests that optimizing Eq (4) is a Pareto optimization problem**.
>     - $\text{(Theorem 4.2)} \qquad \mathbb{E}\_{\tau \sim \pi} \left[ R(\tau) \right] \leq \text{KL} \left[ \pi(\tau) || \beta(\tau) \right] \ \overset{\pi \rightarrow \pi^{\*}}{\Longrightarrow} \ \mathbb{E}\_{\tau^{\*} \sim \pi^{\*}} [ R(\tau) ] = - \mathbb{E}\_{\tau^{\*} \sim \pi^{\*}} [ \ln \beta(\tau)] \ \Longleftrightarrow \ R(\tau)=-\ln\beta(\tau) , \ \ \forall{\tau^{\*}} \sim \pi^{\*}$
> ---
> 4. **Both Eq (6) and Theorem 4.2 describe the optimal state of Eq (4)**, which implies a Pareto optimal solution $\tau^{\*} \sim \pi^{\*}$. This means that **Eq (6) and the result of Theorem 4.2 is just describing the same phenomenon**, i.e., the Pareto optimal state, **from the different perspective**, **one focusing on bi-objectiveness** (Eq (6)) **and the other on the trade-off** **between the two objectives** (Theorem 4.2).
>     - $\text{Eq (6)} \qquad \pi^{\*}(\tau)=\frac{\beta(\tau)e^{R(\tau)}}{e^{1-\lambda}} = \beta(\tau)e^{R(\tau)} \quad \left( \because \lambda = 1 \right)$
> ---
> 5. In a Pareto optimization problem, it is obvious that **any policy that does not achieve Pareto optimality**, i.e., $\pi(\tau)\neq \beta(\tau)e^{R(\tau)}$, **always has room for Pareto improvement**. And we know that Pareto optimality presented in Theorem 4.2 is derived from RUBO presented in Theorem 4.1.
> ---
> 6. Here, through *reductio ad absurdum* (i.e., through the proof by contradiction), **Theorem 4.3 shows that** the contraposition of Theorem 4.1, i.e., $\mathbb{E}\_{\tau \sim \pi}[R(\tau)] > \text{KL}[\pi(\tau)||\beta(\tau)]$, leads to $\pi(\tau)\neq \beta(\tau)e^{R(\tau)}$, or **the negation of Theorem 4.1 leads to the negation of Eq (6), and Pareto optimality does not hold.** In other words, **Theorem 4.3 provides the condition for the Pareto improvement.**
>     - $\text{(Theorem 4.3)} \qquad \mathbb{E}\_{\tau \sim \pi}[R(\tau)] > \text{KL}[\pi(\tau)||\beta(\tau)]  \ \overset{\text{Eq (14)}}{\Longrightarrow} \ \mathbb{E}\_{\tau \sim \pi}[R(\tau)] + \mathbb{E}\_{\tau \sim \pi}[\ln \beta(\tau)] > 0 \ \Longrightarrow \ \pi(\tau)\neq \beta(\tau)e^{R(\tau)}$
> ---
> 7. Therefore, **by forcing the Pareto improvement condition given by Theorem 4.3 to hold**, i.e.,  $\mathbb{E}\_{\tau \sim \pi}[R(\tau)] > \text{KL}[\pi(\tau)||\beta(\tau)] \ \Longrightarrow \ \mathbb{E}\_{\tau \sim \pi}[R(\tau)] + \mathbb{E}\_{\tau \sim \pi}[\ln \beta(\tau)] > 0$, **we can ensure that the target policy $\pi$ or the target LM $\pi_{\theta}$ is updated only in the direction of Pareto improvement** that increases both the likelihood and reward objectives without unnecessary exploration, and as a result reach the Pareto optimal solution faster.
> ---
> 8. **The message behind Theorem 4.3** **is simple**: **The Pareto improvement condition is satisfied when $\mathbb{E}\_{\tau \sim \pi}[R(\tau)] + \mathbb{E}\_{\tau \sim \pi}[\ln \beta(\tau)] > 0$, so manipulate either or both $R(\cdot)$ and $\beta(\cdot)$ to hold it.**
> ---
> 9. Given that the behavior policy is either a pre-defined distribution, i.e., $\beta(\cdot)$, or a pre-trained LM, i.e., $\beta_{\bar{\theta}}(\cdot)$, whose parameter is fixed as $\bar{\theta}$, it is only $R(\cdot)$ that can be manipulated. Accordingly, **we only need to focus on $R(\tau)$ to hold Eq (7)**. Specifically, we need to **manipulate $R(\tau)$ such that only a few high rewards are considered** because $\mathbb{E}\_{\tau \sim \pi} \left[ R(\tau) \right]$ refers to the average reward $\tilde{r}$ and the average is sensitive to bias caused by outliers. Extremely saying, **a single high reward is more effective at satisfying the Pareto improvement condition**, i.e., $\mathbb{E}\_{\tau \sim \pi}[R(\tau)] + \mathbb{E}\_{\tau \sim \pi}[\ln \beta(\tau)] > 0$, **than many average rewards.**
> ---
> 10. **This is how we came to propose Reward Dropout based on Theorems 4.1, 4.2, and 4.3.**
> ---
> - Reviewer #1 and Reviewer #3 have raised some questions related to this response. We sincerely appreciate the reviewers for their thoughtful feedback.

---

> ### Author Response · Authors · 2023-11-20
>
> # MR3) Objective Function of RLM, Bi-objective Problem, and Off-policy RL.
>
> ---
>
> - Some reviewers have asked two insightful questions regarding the objective function of RLM, Eq (4), or equivalently Eq (5), and its solution Eq (6). The questions are as follows:
>     - Q1) *"In Eq (4), why is the reward function exponentiated, and why does this make sense in the context of the problem?" (by reviewer #1)*
>     - Q2) *" According to the experimental results such as Fig. 3, if you only consider the reward metric and would like to relatively neglect the likelihood objective, this weakens the motivation for using bi-objective formulation. Then, why not consider the trade-off problem from the perspective of constrained optimization with the likelihood objective as the constraint?" (by reviewer #3)*
> - After reading the questions carefully and crystallizing their essence, we ultimately concluded that the essence of the question was:
>     - Q1) *“Why should the objective function of RLM in our study be Eq (4), i.e. Eq (5)?”*
>     - Q2) *"Since the off-policy RL approach fixes the behavior policy and thus does not seem to maximize the likelihood objective, does it solve the bi-objective problem described in Eq (6)? i.e., does off-policy RL address bi-objective optimization, rather than constrained optimization?”*
> - We think these are very thought-provoking questions to help understand our research, so we decided to share the answers to them as meta-responses.

---

> ### Author Response · Authors · 2023-11-20
>
> # MR3) Objective Function of RLM, Bi-objective Problem, and Off-policy RL (cont.).
>
> ---
>
> - To answer the first question, ****we first need to clarify the objective function used to train the model in RLM studies. In general**, RLM studies maximize the objective function defined below to train RLMs:**
>     - $\text{(Objective Function of RLM)} \qquad \mathbb{E}\_{\tau \sim \pi_{\phi}^{\text{RL}}} [R(\tau)] - \alpha\text{KL}[\pi_{\phi}^{\text{RL}}(\tau)||\pi^{\text{SFT}}(\tau)]$
>         - $\pi_{\phi}^{\text{RL}}$ is the target model parameterized by $\phi$  (= target policy $\pi(\tau)$) and optimized for a reward $R(\tau)$ via reinforcement learning.
>         - $\pi^{\text{SFT}}$ is a pre-trained behavior model (= behavior policy $\beta(\tau)$) fine-tuned on the pretraining data $\mathcal{D}\_{\text{pretrain}}$ within a supervised learning manner.
>         - $\alpha$ is a user-defined parameter that balances how much to consider reward maximization versus likelihood maximization (i.e., maximizing convergence between the target model and the bebavior model). It is often customized by the user, but it is also possible to estimate $\alpha$ adaptively based on meta-learning.
>     - Please refer to the following literature to check how RLM studies define the objective function.
>         - [Stiennon et al. (2020)](https://proceedings.neurips.cc/paper/2020/file/1f89885d556929e98d3ef9b86448f951-Paper.pdf), page 6, Section 3.4, Human Feedback Policies paragraph
>         - [Korbak et al. (2022)](https://aclanthology.org/2022.findings-emnlp.77.pdf), Section 3, Equation (4)
>         - [Perez et al. (2022)](https://arxiv.org/pdf/2202.03286.pdf), Section 2.2, Reinforcement Learning paragraph
>         - [Korbak et al. (2022)](https://proceedings.neurips.cc/paper_files/paper/2022/file/67496dfa96afddab795530cc7c69b57a-Paper-Conference.pdf), page 3, Equation (2)
>         - [Ouyang et al. (2022)](https://proceedings.neurips.cc/paper_files/paper/2022/file/b1efde53be364a73914f58805a001731-Paper-Conference.pdf)’s [Supplementary Material](https://proceedings.neurips.cc/paper_files/paper/2022/file/b1efde53be364a73914f58805a001731-Supplemental-Conference.pdf), page 31, Section D.4, Equation (2)
>         - [Bai et al. (2022)](https://arxiv.org/pdf/2204.05862.pdf), page 17, Equation (4.1)
> ---
> - We can see that Eq (4) in our paper is a dual problem of maximizing the RLM objective function when $\alpha=1$.
>     - $\text{(Objective Function of RLM)} \qquad \mathbb{E}\_{\tau \sim \pi_{\phi}^{\text{RL}}} [R(\tau)] - \alpha\text{KL}[\pi_{\phi}^{\text{RL}}(\tau)||\pi^{\text{SFT}}(\tau)] \quad \overset{\times -1}{\underset{\alpha=1}{\Longrightarrow}} \quad \text{KL}[\pi_{\phi}^{\text{RL}}(\tau)||\pi^{\text{SFT}}(\tau)] - \mathbb{E}\_{\tau \sim \pi_{\phi}^{\text{RL}}}[R(\tau)] = \sum_{\tau} \pi_{\phi}^{\text{RL}}(\tau) \ln \frac{\pi_{\phi}^{\text{RL}}(\tau)}{\pi^{\text{SFT}}(\tau)e^{R(\tau)}}$
>     - $\text{Eq (4)} \qquad \text{KL}\left[ \pi(\tau) || \beta(\tau) e^{R(\tau)} \right] = \sum_{\tau} \pi(\tau) \ln\frac{\pi(\tau)}{\beta(\tau)e^{R(\tau)}}$
>     - That is, Eq (4) is the special case of RLM objective where $\pi^{\text{RL}}\_{\phi}(\tau) = \pi(\tau)$, $\pi^{\text{SFT}}(\tau) = \beta(\tau)$ and $\alpha=1$.
>     - It is reasonable to have $\alpha=1$ in the sense that we value reward maximization and KL divergence minimization in equal weights, i.e., we consider a balance between the two objectives.
> ---
> - Now we can provide the answer for the first question *"Q1) In Eq (4), why is the reward function exponentialized, and why does this make sense in the context of the problem?"*. The answer is **"because that is how the objective function of the RLM is defined"**. Of course, there are several known benefits to using exponentiated rewards that are independent of the RLM's objective function. For example, **by defining an exponentiated reward**, we can **impose an optimistic bias that encourages an agent to strongly pursue higher rewards**. Also, exponentiation guarantees the reward to be a positive real number, and thus **eliminates the instability of training caused by zero or negative rewards**. Nevertheless, we prefer to answer that the exponentiated reward is naturally induced by the definition of the RLM objective function.

---

> ### Author Response · Authors · 2023-11-20
>
> # MR3) Objective Function of RLM, Bi-objective Problem, and Off-policy RL (cont.).
>
> ---
>
>
> - The second question *“why not consider the trade-off problem from the perspective of constrained optimization with the likelihood objective as the constraint?”* asks whether “*off-policy RL addresses bi-objective optimization not constrained optimization?”* To answer this question, we first need to point out that **maximizing the objective function of RLM can be solved essentially as a constrained optimization problem**, **i.e., the problem of training an RLM is equivalent to a constrained optimization problem**:
>     - $\argmax_{\phi} \mathbb{E}\_{\tau \sim \pi_{\phi}^{\text{RL}}} [R(\tau)] \quad \textit{s.t.} \quad \text{KL}[\pi_{\phi}^{\text{RL}}(\tau)||\pi^{\text{SFT}}(\tau)] = 0 \qquad \Longrightarrow \qquad \mathbb{E}\_{\tau \sim \pi_{\phi}^{\text{RL}}} [R(\tau)] - \alpha\text{KL}[\pi_{\phi}^{\text{RL}}(\tau)||\pi^{\text{SFT}}(\tau)]$
>     - Using the Lagrange method, we can convert the constrained problem above into an unconstrained problem with $\alpha$ as a Lagrange multiplier.
>     - The unconstrained problem is equivalent to the problem of maximizing RLM’s objective function of RLM shown above.
> ---
> - For the answer to the first question, we mentioned that Eq (4) in our paper is equivalent to the objective function of the RLM when $\alpha=1$. **Therefore, minimizing Eq (4), or solving Eq (5), is nothing but to solve a constrained optimization problem where the objective function and constraints are considered with equal weight ($\alpha=1$)**. In other words, the bi-objective optimization problem is a case of a constrained optimization problem where the objective function and constraints are given equal weight, so we don't need to explicitly consider a constrained optimization perspective with likelihood objectives as constraints.
> ---
> - Now let us explain why, even though it is obvious that Eq (4) deals with a bi-objective optimization problem (as a case of a constrained optimization problem where the objective function and constraints are given equal weight), the result "*seems to optimize only the reward metric, neglecting the likelihood objective, as shown in Fig 3"*. To conclude, **the reason why Fig. 3 looks like a single-objective optimization is that when we implement the off-policy RLM training, we follow the practical implementation that is preferred in the RLM research.**
>     - First, let us show mathematically how the off-policy RLM is implemented and whether it achieves bi-objective optimization. To minimize Eq (4) in the context of off-policy RL, we can implement the off-policy policy gradient (off-PG) algorithm as follows:
>         - $\text{(Reward Only Objective Function)} \qquad J(\pi_{\theta})=\mathbb{E}\_{\tau \sim \pi_{\theta}}[R(\tau)]$
>         - $\text{(Off-PG)} \qquad \nabla_{\theta}J(\pi_{\theta})= \sum_{\tau} \nabla_{\theta}\pi_{\theta}(\tau) R(\tau) = \sum_{\tau} \beta_{\bar{\theta}}(\tau) \frac{\pi_{\theta}(\tau)}{\beta_{\bar{\theta}}(\tau)}\frac{\nabla_{\theta}\pi_{\theta}(\tau)}{\pi_{\theta}(\tau)} R(\tau) = \mathbb{E}\_{\tau \sim \beta_{\bar{\theta}}}\left[ R(\tau)\bigg(\frac{\pi_{\theta}(\tau)}{\beta_{\bar{\theta}}(\tau)}\bigg) \nabla_{\theta}\ln\pi_{\theta}(\tau) \right]$
>         - Here, $\frac{\pi_{\theta}(\tau)}{\beta_{\bar{\theta}}(\tau)}$ is the importance score. The higher the probability that the behavior trajectory $\tau \sim \beta_{\bar{\theta}}$ is under the target policy distribution $\pi_{\theta}(\cdot)$, the higher the importance score. The behavior trajectory $\tau \sim \beta_{\bar{\theta}}$ will have a very high probability under the behavior policy distribution $\beta_{\bar{\theta}}$. The importance score implies that the target policy parameter $\theta$ is updated such that $\pi_{\theta}$ has at least a high probability as $\beta_{\bar{\theta}}$ for all the behavior trajectories $\forall\tau \sim \beta_{\bar{\theta}}$.
>         - In summary, Off-PG forces $\pi_{\theta}$ to converge to $\beta_{\bar{\theta}}$, i.e., while maximizing $R(\tau)$, it also maximizes $\beta_{\bar{\theta}}(\tau)$. This is equivalent to solving the bi-objective problem represented by Eq (4). Thus, we can say that off-policy RLM explicitly performs bi-objective optimization.

---

> > ### Author Response · Authors · 2023-11-20
> >
> > # MR3) Objective Function of RLM, Bi-objective Problem, and Off-policy RL (cont.).
> > ---
> > - We are now convinced that **an off-policy RLM implemented with Off-PG maximizes the reward objective and the likelihood objective simultaneously.** Accordingly, if we visualize the trajectories of rewards and likelihoods of an off-policy RLM under training, we will see that **rewards and likelihoods increase simultaneously up to a certain level, after which rewards can no longer increase due to the RUBO given in Theorem 4.1.** If training continues after reward touched RUBO, then RUBO will decrease as $\pi_{\theta}$ gets close to $\beta_{\bar{\theta}}$, that is, $\text{KL}\left[ \pi_{\theta}(\tau) || \beta_{\bar{\theta}}(\tau) \right]$ will converge to zero, resulting in the expected reward $\mathbb{E}\_{\tau \sim \pi{\theta}} \left[ R(\tau) \right]$ will also converge to zero. This means $\pi_{\theta}(\tau) \rightarrow \beta_{\bar{\theta}}(\tau)$, or $\frac{\pi_{\theta}(\tau)}{\beta_{\bar{\theta}}(\tau)} \rightarrow 1$, and it will lead to $\mathbb{E}\_{\tau \sim \pi_{\theta}} \left[ R(\tau) \right] = \text{KL} \left[ \pi_{\theta}(\tau) || \beta_{\bar{\theta}}(\tau) \right]  \rightarrow 0$ and $\mathbb{E}\_{\tau \sim \pi_{\theta}} \left[ R(\tau) \right] \approx \mathbb{E}\_{\tau \sim \beta_{\bar{\theta}}} \left[R(\tau) \right]$. Here, $\mathbb{E}\_{\tau \sim \pi_{\theta}} \left[ R(\tau) \right] \approx \mathbb{E}\_{\tau \sim \beta_{\bar{\theta}}} \left[R(\tau) \right]$ implies the theoretical issue that **if the training time goes to infinite, theoretically, the expected reward of $\pi_{\theta}$ will get to depend entirely on $\beta_{\bar{\theta}}$.**
> >     - $\text{(Theorem 4.1)} \qquad \mathbb{E}\_{\tau \sim \pi_{\theta}} \left[ R(\tau) \right] \leq \text{KL} \left[ \pi_{\theta}(\tau) || \beta_{\bar{\theta}}(\tau) \right] = \text{RUBO}$
> >     - $\text{(Theoretical Issue)} \qquad \mathbb{E}\_{\tau \sim \pi_{\theta}} \left[ R(\tau) \right]  = \mathbb{E}\_{\tau \sim \beta_{\bar{\theta}}} \left[ \left( \frac{\pi_{\theta}(\tau)}{\beta_{\bar{\theta}}(\tau)} \right) R(\tau) \right] \approx \mathbb{E}\_{\tau \sim \beta_{\bar{\theta}}} \left[R(\tau) \right] \quad \left(\because \ \frac{\pi_{\theta}(\tau)}{\beta_{\bar{\theta}}(\tau)}\rightarrow1 \right)$
> >     - We experimented and validated that the reward increases and then decreases after touching the RUBO. Please refer to Appendix I.

---

> ### Author Response · Authors · 2023-11-20
>
> # MR3) Objective Function of RLM, Bi-objective Problem, and Off-policy RL (cont.).
>
> ---
>
> - Based on the theoretical issues discussed above, we can explain why we provided Fig. 3 which seems to weaken bi-objective motivation. The reasons are as follows.
>     - **To circumvent the theoretical issues mentioned above, RLM researchers devised a trick: transfer learning + on-policy RL.**
>         - **First, we initialize the parameter $\theta$ of $\pi_{\theta}$ to the parameter $\bar{\theta}$ of $\beta_{\bar{\theta}}$, the pre-trained large language model.** **This is the same as the transfer learning approach** in that the target model’s parameters are initialized to those of the pre-trained giant model.
>             - For example, in the Objective Function of RLM introduced earlier, we replace $\phi$ of $\pi^{\text{RL}}\_{\phi}$ with the parameters of $\pi^{\text{SFT}}$.
>         - **Then, maximize the Reward Only Objective Function, i.e., $\mathbb{E}\_{\tau \sim \pi_{\theta}} \left[ R(\tau) \right]$ over a target trajectory $\tau \sim \pi_{\theta}$, where $\pi_{\theta}$ is initialized with behavior parameter $\bar{\theta}$**. In other words, set the initial distribution of $\pi_{\theta}$ to $\beta_{\bar{\theta}}$, and update $\theta$ along with $\hat{\theta} \leftarrow \theta + \alpha R(\tau)\nabla_{\theta} \ln \pi_{\theta}(\tau)$. **It is identical to the on-policy RL approach** in that it directly maximizes $\mathbb{E}\_{\tau \sim \pi_{\theta}} \left[ R(\tau) \right]$ with target trajectories rather than behavior trajectories.
>
>     - **If we implement RLM via transfer learning + on-policy RL, this is a single-objective problem that maximizes only $R(\cdot)$.** Therefore, as training continues, the target policy $\pi_{\theta}$ is forced to diverge from $\beta_{\bar{\theta}}$, i.e., minimize the likelihood objective $\beta_{\bar{\theta}}(\cdot)$, in order to maximize the reward objective $R(\cdot)$ only.
>     - **However**, **if we initialize $\pi\_{\theta}$ with the large language model $\beta_{\bar{\theta}}$ whose parameters are robust enough against any parameter update**, it is empirically known that no matter how much we optimize $\theta$ with respect to $R(\tau)$, **$\pi\_{\theta}$ does not diverge from $\beta\_{\bar{\theta}}$.**
>     - **Hence, the transfer learning + on-policy RL trick has become a common practice in RLM research for practical reasons.** A large number of RLM studies follow this trick, including the RLM literature I mentioned earlier.
>     - **Our work borrowed this trick as well.** Specifically, **we applied the transfer learning part** to initialize the target policy with the behavior policy, i.e., a pretrained behavior LLM. **On the other hand**, considering that the purpose of this study is to analyze an RLM from a bi-objective perspective, **we discarded the on-policy RL part**, as it reduces RLM training to a single-objective problem. **Instead, we implemented the Off-PG as an update algorithm for RLMm, maintaining the context of off-policy RL.**
>     - Please refer to the Algorithm Table in Appendix D.4. for the details.
>     - We mentioned that the reason why **RLM researchers begin to use the transfer learning + on-policy RL trick is because it allows them to improve $R(\tau)$ without $\pi_{\theta}$ diverging from $\beta_{\bar{\theta}}$.** **This effect is even stronger in our algorithm that replaces on-policy RL with off-policy RL**. **Therefore, we thought that it is sufficient to visualize the change in $R(\tau)$ as training continues and that it would be more impressive.** This is why we provided Fig. 3 which seems to weaken bi-objective motivation.
> ---
> - To summarize:
>     1. Due to the theoretical issue, we applied a trick commonly used in RLM research with a slight variation (transfer learning + off-policy RL) to our study.
>     2. It is a well-known fact that $\pi\_{\theta}$ does not diverge easily from $\beta_{\bar{\theta}}$ when using this trick.
>     3. Therefore, we decided that it is sufficient to show the trend of change for $R(\tau)$ only, and thus provided a plot like Fig 3.
> ---
> - Reviewer #1 and Reviewer #3 have raised some questions related to this response. We sincerely appreciate to the reviewers for their thoughtful feedback.

---

### Meta-Review · Area_Chair_uH5F · 2023-12-08

**Metareview:**

This paper proposes theory to help understand controllable language models (LMs). While the authors do a commendable job of surveying the literature of controllable LMs and unifying the exposure, unfortunately the theory that is put forth in this paper is either thin or incorrect, which leads to the decision to reject the paper. Here are a few examples where the reviewers and the AC were not convinced that the proposed theory is adding to the published literature:

- In the proof of Thm 4.1(page 14), the first few lines are dedicated to proving non-negativity of KL divergence, which is a well-known fact. Why is this being reproven? The assumption that $\pi(\tau) = \beta(\tau) e^{R(\tau)}$ is too strong. It is basically assuming that the reward is normalized in a way that $\beta(\tau) e^{R(\tau)}$ becomes a valid distribution. And with that assumption the claim is immediate. The last line of the proof of Theorem 4.1 is not even understandable; as it is not clear what $E [\pi(\tau) || e^{R(\tau)}]$ is. These issues aside, it is not clear to the AC and the reviewers what the significance of the claim is.

- Theorem 4.2 (page 15) cannot be correct as stated. Let $R(\tau) = -\log \beta(\tau)$. Then, $\pi(\tau) = \beta(\tau)  e^{R(\tau)} = 1,$ which is not a valid distribution as it doesn't sum to $1$. In the first proof attempt, the issue is that the policy is considered to concentrate on a single outcome, which is not true because the optimal policy is $\beta(\tau) e^{R(\tau)},$ whose entropy is generally non-zero. In the second proof attempt, the problem arises right after (13) where $\log \pi(\tau)$ is ignored. These issues aside, it is not clear what the significance of this claim is.

Given the serious issues with the theoretical framing of this paper, unfortunately the decision must be to reject the paper.

**Justification For Why Not Higher Score:**

- Serious issues with the theoretical framing of the paper.

**Justification For Why Not Lower Score:**

- The literature survey of the paper on controllable LM framing is commendable, and could be a useful blogpost already.

---

### Decision · Program_Chairs · 2024-01-16

Reject